



# Modelling the impacts of emission changes on $O_3$ sensitivity, atmospheric oxidation capacity and pollution transport over the Catalonia region

Alba Badia[1], Veronica Vidal[1,2], Sergi Ventura[1], Roger Curcoll[3], Ricard Segura[1], and Gara Villalba[1,4]

[1]Sostenipra Research Group, Institute of Environmental Sciences and Technology, Universitat Autònoma de Barcelona, 08193 Bellaterra, Barcelona, Spain.
[2]Departament d'Arquitectura de Computadors i Sistemes Operatius (CAOS), Escola d'Enginyeria, Universitat Autònoma de Barcelona 08193 Bellaterra, Barcelona, Spain.
[3]Institut de Tècniques Energètiques (INTE), Universitat Politècnica de Catalunya, Barcelona, Spain
[4]Department of Chemical, Biological and Environmental Engineering, Universitat Autònoma de Barcelona, 08193 Bellaterra, Barcelona, Spain

**Correspondence:** alba.badia@uab.cat

**Abstract.** Tropospheric ozone ($O_3$) is an important surface pollutant in urban areas, and it has complex formation mechanisms that depend on the atmospheric chemistry and meteorological factors. The severe reductions observed in anthropogenic emissions during the COVID-19 pandemic can further our understanding of the photochemical mechanisms leading to $O_3$ formation and provide guidance for policies aimed at reducing air pollution. In this study, we use the air quality model WRF-Chem coupled with the urban canopy model BEP-BEM to investigate changes in the ozone chemistry over the Metropolitan Area of Barcelona (AMB) and its atmospheric plume moving northwards, which is responsible for the highest number of hourly $O_3$ exceedances in Spain. The trajectories of the air masses from the AMB to the Pyrenees are studied with the Lagrangian particle dispersion model FLEXPART-WRF. The aim is to investigate the response of ozone chemistry to changes in the precursor emissions. The results show that with the reduction in emissions: 1) the ozone chemistry tends to enter the NOx-limited or transition regimes; however, highly polluted urban areas are still in the VOC-limited regime, 2) the reduced $O_3$ production is overwhelmed by reduced nitric oxide (NO) titration, resulting in a net increase in the $O_3$ concentration (up to 20%) in the evening, 3) the increase in the maximum $O_3$ level (up to 6%) during the lockdown could be attributable to an enhancement in the atmospheric oxidation capacity (AOC), 4) the daily maximum levels of ozone and odd oxygen species ($O_x$) generally decreased (4%) in May with the reduced AOC, indicating an improvement in the air quality, and, 5) ozone precursor concentration changes in the AMB contribute to the pollution plume moving along the S–N valley to the Pyrenees. Our results indicate that $O_3$ abatement strategies cannot rely only on $NO_x$ emission control but must include a significant reduction in anthropogenic sources of VOCs. In addition, our results show that mitigation strategies intended to reduce $O_3$ should be designed according to the local meteorology, air transport, particular ozone regimes and AOC of the urban area.



# 1 Introduction

Tropospheric ozone ($O_3$) is a radiatively active gas that acts as an oxidizing agent and a surface pollutant in urban areas, where it is a major component of photochemical smog and causes a number of respiratory health effects (Sillman, 2003; Anenberg et al., 2010). In 2019, it was estimated that 365,000 respiratory mortalities worldwide were due to anthropogenic $O_3$, representing an increase of $\sim 16\%$ with respect to 2010 (GBD 2019 Risk Factors Collaborators, 2020). The World Health Organization (WHO) provides guidelines to protect humans from exposure by reducing the levels of key air pollutants, including ozone. The WHO released revised air quality guidelines in 2021, keeping the $O_3$ levels to 100 $\mu$g m-3 for an 8-hour daily average and recommending 60 $\mu$g m$^{-3}$ for an 8-hour daily average during the six consecutive months with the highest average ozone concentrations (Organization, 2021).

Ozone is photochemically produced through nonlinear chemical processes, involving mainly reactions of nitrogen oxides ($NO_x$= $NO_2$+NO) and volatile organic compounds (VOCs); this results from both anthropogenic and biogenic sources in the presence of sunlight (Monks et al., 2015; Crutzen, 1974; Derwent et al., 1996). This chemistry occurs in two photochemical regimes, NOx-sensitive and VOC-sensitive (Sillman et al., 1990; Sillman, 1999). In the NOx-sensitive regime (low NOx and high VOC), ozone production is controlled (or limited) by the concentration of $NO_x$, therefore, the $O_3$ levels increase with increasing $NO_x$, and there are only small changes with increases in the VOC levels. In this regime, $O_3$ reacts mainly with hydrogenated species to form hydrogen peroxide ($H_2O_2$), which is removed by wet and dry deposition. In the VOC-sensitive regime (high $NO_x$), ozone levels increase with increased VOC concentrations and decrease with increased $NO_x$ concentrations. This last regime is typical of urban areas where a reduction in $NO_x$ emissions enhances ozone levels locally due to the higher levels of oxidants (mainly hydroxyl radicals, OH) and reactions with VOCs. Peak concentrations of ozone usually occur during the midday hours, when the sunlight is most intense. However, during the afternoon and evening, high ozone concentrations are observed in remote areas due to higher biogenic VOC emissions, less titration by NO, and transport of $O_3$ and its precursors from their sources. At night and next to a source with high emissions of NO (e.g., power plants), ozone is lost through the process of $NO_x$ titration and forms $NO_2$, which is subsequently converted to nitric acid ($HNO_3$) and removed from the atmosphere by wet and dry deposition (Monks et al., 2015). In summary, $O_3$ levels can be reduced only if there are reductions in the amounts of the precursors $NO_x$, VOC and carbon monoxide (CO). Reductions in VOC emissions would be an effective pathway to reducing ozone in a high $NO_x$ area (VOC-sensitive). On the other hand, reductions in $NO_x$ emissions would be effective in reducing $O_3$ if $NO_x$-sensitive chemistry dominates. In urban areas, this photochemistry illustrates the difficulties involved in developing policies to reduce $O_3$ in polluted regions (Sillman, 2003). Thus, a more profound understanding of the sensitivity of local ozone formation to changes in $NO_x$ and VOC levels is essential for developing effective air quality policies. Other important processes for removal of $O_3$ are reactions of halogens species (Badia et al., 2021a), which remove 30–35 Tg (11–15 %) of tropospheric ozone, and dry deposition, which accounts for about 20% of the $O_3$ lost from the troposphere (Wild, 2007).

In addition to photochemical reactions, the concentration of ozone is sensitive to meteorological variables such as solar radiation and wind speed and direction (Neiburger, 1969). Ozone production is intensified on warm, sunny days when the





air is stagnant. Therefore, the increase in frequency, severity, and duration of heatwaves during recent decades increases the
need to understand the influence of meteorological drivers and anthropogenic factors on ground-level ozone pollution. This
is becoming more important for urban cities in the Mediterranean area with high summer temperatures, and heatwaves are
projected to become more severe in the future due to anthropogenic climate change (Pyrgou et al., 2018; Zittis et al., 2015).

Indeed, local ozone levels depend not only on local production and loss mechanisms that are sensitive to meteorological
factors but also on the transport of ozone and its precursors. Previous studies have shown that ambient ozone concentrations
are strongly influenced by transport of regional ozone and its precursors, while local precursor emissions play limited roles
in ozone formation (Romero-Alvarez et al., 2022; Kleanthous et al., 2014). Cristofanelli and Bonasoni (2009) showed that
the background tropospheric ozone concentration in the Mediterranean area and southern Europe is affected mainly by three
transport processes: 1) regional and long-range transport of pollutants, 2) downwards transport from the stratosphere, and 3)
transport of dust from the Sahara Desert.

The lockdown period provided a significant reduction in ozone precursors, and it represents an excellent opportunity to
further our understanding of the photochemical reactions involved in ozone chemistry. Estimated average emission reductions
in Spain during the most severe lockdown period were reported, with road and air traffic reductions reaching 80-90% Guevara
et al. (2021). During this period, Guevara et al. (2021) estimated the average emission reductions at the EU-30 level to be
-33% for $NO_x$. Consequently, restrictive mobility measures that included important reductions in traffic had many positive
environmental impacts, 70 and improvements in air quality were reported globally (Liu et al., 2020; Venter et al., 2020a;
Sharma et al., 2020).

Previous modelling studies analysed the changes in air quality, emissions, and chemical regimes seen on global and regional
scales during the COVID-19 lockdown (Miyazaki et al., 2021; von Schneidemesser et al., 2021; Roozitalab et al., 2022; Sicard
et al., 2020; Badia et al., 2021). A global modelling study by Miyazaki et al. (2021), showed that the global total tropospheric
ozone burden declined by 6 Tg (~2%) during May-June 2020, mainly due to emission reductions in Asia and the Americas.
The modelling study of Venter et al. (2020b) found that, after taking into account the meteorological variations, lockdown
measures have reduced the levels of $NO_2$, mainly due to the reduction in transportation, and PM levels by approximately 60%
and 31% in 34 countries, with a general increase in $O_3$ of 4% (-2 to 10%). Sicard et al. (2020) described ozone increases in
cities (17% in Europe, 36% in Wuhan) resulting from lower titration of $O_3$ by NO due to the strong reduction in $NO_x$ emissions
from road transport. Most of this literature was focused only on the lockdown period, and the de-escalation period, which had
different ozone chemistry, was not analysed. In addition, these studies did not discuss changes in the chemical regimes arising
for different land uses and the transport of pollutants due to lockdown measures from cities to rural areas.

Only a few studies have reported that enhanced atmospheric oxidation capacity (AOC) contributed to $O_3$ increases during
the COVID-19 lockdown due to increases in the major oxidants OH, hydroperoxy radical ($HO_2$) and nitrate radical ($NO_3$) (Zhu
et al., 2021; Wang et al., 2021b, 2022). The predominant oxidant for AOC during the daytime is OH, which is responsible for
the oxidation and removal of most natural and anthropogenic trace gases (Elshorbany et al., 2009; Saiz-Lopez et al., 2017).
During the night, the concentration of OH is significantly reduced, and the AOC is then controlled by $NO_3$, together with $O_3$,
which is also an important oxidant (Elshorbany et al., 2009; Saiz-Lopez et al., 2017). During the lockdown, the significant



decreases in NO$_2$ concentrations increased the OH levels, which led to the formation of harmful oxidants such as O$_3$ (Zhu

et al., 2021; Wang et al., 2021b, 2022).

In recent decades, EU emission mitigation policies have been successful in decreasing emissions of key air pollutants such as SO$_2$, NO$_x$, NMVOCs, and PM (Sicard et al., 2021; and European Environment Agency et al., 2019). However, the current levels for the secondary air pollutant O$_3$ in cities continue to exceed the EU standards and WHO air quality guidelines (Guerreiro et al., 2014; and European Environment Agency et al., 2019). Indeed, local ozone pollution mitigation efforts are generally

inefficient, mainly because 1) ozone formation depends on nonlinear chemical interactions, 2) with a lifetime of several weeks, ozone levels are strongly influenced by long-distance transport, which is associated with specific weather conditions and the hemispheric background, and 3) its precursors are emitted mostly far from the sites of ozone exceedances. Sicard et al. (2013) found a decrease in annual O$_3$ averages (-0.4% yr-1) at rural sites and an increase at urban and suburban stations (0.6% and 0.4%, respectively) during 2000–2010 for the Mediterranean area. These changes resulted from the mitigation policies

for NO$_x$ and VOC emissions in the EU that led to an increase in O$_3$ levels in urban areas due to a reduction in titration by NO. Therefore, there is an urgent need to expand our knowledge of ozone chemistry to help decision-makers choose better mitigation strategies.

In the present study, we use the air quality model WRF-Chem coupled with the urban canopy model BEP-BEM to investigate the response of NO$_x$-VOCs-O$_3$ chemistry to changes in precursor emissions in the Metropolitan Area of Barcelona (AMB); this

furthers our understanding of ozone formation mechanisms and transport to rural areas, and it enables the design of effective air quality mitigation strategies. We compare the ozone levels for two periods with very different anthropogenic and biogenic activity levels: March and April 2020, when ozone precursors were at their lowest levels because of repeated lockdowns due to COVID, and 2) May 2020, when the AMB was in a de-escalation plan but the ozone levels were typically highest because of the bloom of biogenic emissions under the intense sunlight conditions. In addition, ozone production regimes for different land uses

of the AMB are analysed to determine changes in the NOx/VOC ratio in different areas of the city that are affected by different biogeochemical effects (biogenic VOC emissions, dry deposition, anthropogenic emissions of ozone precursors). Here, we also discuss changes in the AOC and propose that the AOC should be considered when designing air quality control policies. Changes in ozone circulation from the AMB to the Pyrenees mountains are also discussed for specific days characterized by high ozone levels.

The case study is described in Sect. 2. The air quality model, including the model setup, validation, and simulations, is described in Sect. 3, followed by the results (Sect. 4). A discussion of the ozone chemistry that includes ozone sensitivity, AOC and transport from the city to rural areas is presented in Sec. 5.

## 2    Case study

The Metropolitan Area of Barcelona (AMB) serves as our case study. It is located in Catalonia (Spain) in the northeastern part

of the Iberian Peninsula (Fig. 1). This region is characterized by a Mediterranean climate, with dry and hot summers and clear skies. Due to a complex orographic territory with different altitudes and several peripheral mountain ranges and depressions, it





is not easy to generalize the climatic features of the Catalan lands for the whole territory (Martín-Vide et al., 2010). The AMB, with more than 3 million people, is the most populated urban area on the Mediterranean coast.

The city of Barcelona annually reports some of the highest air pollution levels in Europe, and the most problematic pollutants
are $NO_2$, PM2.5, and PM10 (Rivas et al., 2014). In particular, in 2019, the $NO_2$ annual mean levels in the high traffic urban air pollution ground monitoring stations (Eixample and Gràcia-Sant Gervasi) exceeded the WHO guideline (40 $\mu g\ m^{-3}$) (Rico et al., 2019). In the same year, the mean values for PM2.5 and PM10 exceeded the WHO guideline (20 and 10 $\mu g\ m^{-3}$, respectively) at all urban stations in the city (Rico et al., 2019). Exceeding these air quality reference levels is associated with significant risks to public health (Organization, 2021; Rivas et al., 2014). The year 2020 was the first year in which the $NO_2$
values in Barcelona remained within the WHO limits (Rico et al., 2020) due to the significant reduction in traffic emissions that resulted from the Spanish government's emergency rule and its lockdown restrictions (see Supplement Fig. S1). Note that 2020 had more rainfall than previous years, which has important implications for the removal of pollutants from the air.

Another air quality problem was found for the Vic plain (see Fig. 1), which records the highest number of exceedances for hourly $O_3$ levels (180 $\mu g\ m^{-3}$) when the sea breeze transports the ozone precursors from AMB inland to this rural plane
(Querol et al., 2017; Massagué et al., 2019; Jaén et al., 2021). During the late spring and summer seasons, the combination of daily upslope winds and sea breezes may cause the intrusion of polluted air masses up to 160 km inland. Thus, the air massed from a polluted area (such as the AMB) can be transported northwards and injected at high altitudes (2000–3000 masl) by the Pyrenean mountain ranges (Querol et al., 2017; Massagué et al., 2019).

Ozone levels were higher compared to previous years in Ciutadella, the background station in Barcelona, during the period
March-June 2020 (see Supplement Fig. S2). In Tona, a rural station located in the Vic Plain and situated 45-70 km north of Barcelona and surrounded by high mountains, and in Pardines, also a rural site situated in the Pyrenees mountains (see Fig. 1), high ozone concentrations were registered and clearly exceeded the WHO limit of 60 $\mu g\ m^{-3}$ for peak seasons. Ozone levels were high in these two rural stations during March-June with higher values for 2015-2019 compared to 2020 due to the reduced emissions in the AMB. Note that for some weeks, the $O_3$ levels were higher in 2020 than in 2015-2019 due to meteorological
conditions that increased the levels of ozone precursors (see Supplemental Figs. S1 and S2).

In this study, we selected two periods to discuss the changes in $O_3$ chemistry: 1) a full lockdown period (30 March-12 April, weeks 3-4 in Table 1), in which we found the highest mobility reduction and 2) a relaxation period (18-30 May, weeks 10-11 in Table 1), when restrictions started to relax and $O_3$ formation increased due to the warm temperatures of the late spring (see Table 1).

The first period was characterized by meteorological dynamism in the Iberian Peninsula. The period began with a high-pressure centre in the northern Atlantic Ocean, which generated a cold and dry continental NE flux over Europe. General precipitation was registered in Catalonia during the first days (March 30 – April 4), caused by a low-pressure centre that moved from the Atlantic to the Mediterranean. These conditions were optimal for generating precipitation in the Mediterranean area of the peninsula. The next days were defined by high pressures over Europe, which generated a more stable meteorology with
isolated precipitation and a warm air mass crossing Europe from the south. During this period, no important fluxes from any directions were detected. At the surface level, the Azores anticyclone became stronger and reached pressures above 1030 hPa





over the Atlantic. The second period had more stability due to a strong anticyclonic ridge covering the SW and centre of Europe, resulting in typical summer weather. Winds from the N and NW were detected NE of Catalonia and in the Ebro Valley, respectively, due to the Pyrenees natural barrier, which modifies the trajectories of superficial winds. Low pressures were found over Italy, which intensified the winds from the north over the east of the peninsula. The last days of this period (May 24–30) registered weak precipitation in the Pyrenees and the area NE of Catalonia, with weak winds from different directions (turning from north to south). This was caused by a strong anticyclone located in northern Europe (over 1030 hPa), which generated an undetermined situation with no effects of high or low pressures in the Peninsula. See Table 1) for a summary of the meteorological conditions for these periods.

In addition, we select two days in the lockdown period (the 3rd and 6th of April) and two days in the relaxation period (the 22nd and 26th of May), during which high ozone concentrations were registered (see Table S1 in the Supplement), to study the changes in the $O_3$ circulation from Barcelona (Ciutadella) to the Pyrenees mountains (Pardines), including the Vic plane (Tona) and Montseny. The mean surface temperatures, pressures at the surface level and accumulated precipitation among other meteorological variables for these days are presented for the four sites (Ciutadella, Montseny, Tona and Pardines) in Table S2 of the Supplement. According to the meteorological data from the Servei Meteorològic de Catalunya Servei Meteorològic de Catalunya (SMC), there was no precipitation during these four days except in the Pyrenees (2.9 mm) on the 22nd of May, the wind intensity was low and the surface temperatures were significantly high for this time of the year during the two days in May.

To supplement this information, Figures 3 and 4 show the trajectories of the air masses arriving at the monitoring stations on the selected days; which were modelled with the Lagrangian particle dispersion model FLEXPART-WRF (Brioude et al., 2013). This version of the Lagrangian model works with the Weather Research Forecasting (WRF) mesoscale meteorological model, with the same parametrization as the WRF-Chem model (see section 3.1). The transport model has been run in backwards mode, which means that what is represented in each plot is the residence time, at each grid cell of the map, for the air masses arriving at each site. Twenty-four-hour back trajectories were calculated for each day at a release time of 16 h and with a grid cell size of 0.03 × 0.03 degrees. The air masses on the 3rd of April and 22 of May were transported from the AMB to rural areas such Montseny and the Vic Plain, and we can see an influence from the bottom layers (0-300 m) and the upper layers (300-2000 m) at the different sites. The air masses on the 6th of April were channelled from the AMB northwards to Montseny, the Vic Plain and the Pyrenees. The air masses on the 26th of May were also transported from the AMB northwards to Montseny, the Vic Plain and the Pyrenees, but the air masses that arrived at the surfaces of these locations had strong local components and a larger influences from the upper layers.

## 3 Air quality model

We used the regional chemistry transport model WRF-Chem (Grell et al., 2005) version 4.1, a highly flexible community model for atmospheric research in which aerosol–radiation–cloud feedback processes are considered. The WRF-Chem model



is widely used for simulations of air pollution episodes (Georgiou et al., 2018; Yegorova et al., 2011) and, in particular, the air
quality over the AMB has been analysed in Badia et al. (2021).

## 3.1 Model set-up

The WRF-Chem model is configured with two domains covering the Iberian Peninsula (D1: 9 km×9 km) and the Catalonia
region (D2: 3 km×3 km) with 45 vertical layers up to 100 hPa (Fig. 2). The meteorological and chemical initial and lat-
eral boundary conditions (IC/BCs) were determined using the ERA5 global model data (Hersbach et al., 2020) and WACCM
(Gettelman et al., 2019), respectively. The HERMESv3 preprocessor tool (Guevara et al., 2019) was used to create the an-
thropogenic emissions files from the CAMS-REG-APv3.1 database (Granier et al., 2019). This emission inventory is based
on data from 2016. Biogenic emissions are computed online from the Model of Emissions of Gases and Aerosols from Na-
ture v2 (MEGAN; Guenther et al. (2012)). For the gas-phase chemical scheme, we used the Regional Acid Deposition Model
(RADM2, Stockwell et al. (1990)), which accounts for 63 chemical species, 21 photolytic reactions and 136 gas-phase reac-
tions. NMVOC oxidation in RADM2 only explicitly treats ethane, ethene, and isoprene species, and all other NMVOCs are
classified as grouped species based on OH reactivity and molecular weight. Thus, the RADM2 gas-phase chemical mechanism
grouped the VOCs into 14 species, such as alkane, alkene, aromatic, and formaldehyde. In WRF-Chem, RADM2 is coupled to
the MADE/SORGAM aerosol module (Ackermann et al., 1998; Schell et al., 2001). RADM2 has been broadly used in studies
of the air quality over Europe (Im et al., 2015; Tuccella et al., 2011)

Here, we used a multilayer layer urban canopy scheme, the building effect parameterization (BEP) coupled with the building
energy model (BEP+BEM, (Salamanca et al., 2011)) to represent the urban areas in our domain; this takes into account the
energy consumed by buildings and the anthropogenic heat, which has been previously validated for the area under study
(Ribeiro et al., 2021; Segura et al., 2021). The local climate zone (LCZ) classification (Stewart and Oke, 2012) is used for the
AMB, which associates specific values of the thermal, radiative and geometric parameters of the buildings and ground into 11
urban classes, which are used by the BEP+BEM urban canopy scheme to compute the heat and momentum fluxes in the urban
areas (see Segura et al. (2021) for more details on the use of LCZ and urban morphology). We performed a spin-up of 1 month.
Table 2 describes the main configuration of the model.

## 3.2 Description of the simulation cases

To better understand the impacts of emission reduction measures on air quality, the WRF-Chem model was utilized to calculate
the changes in $O_3$ chemistry during the COVID lockdown period. We ran two simulations: 1) Business As Usual (BAU)
and COVID for the period of March-June 2020 (see Table 1). The COVID run used the emissions changes provided by the
Barcelona Supercomputing Center (Guevara et al., 2021), which were previously used in other studies (von Schneidemesser
et al., 2021; Brancher, 2021). These emission changes varied per day, country and sector. Figure 5 displays the emission changes
used in this study, and the highest changes were found for the road transport (up to 80%) and aviation (up to 90%) sectors.
Inputs for the other emissions (biogenic, dust, sea-spray) and meteorology used in WRF-Chem were set to be consistent. As





a result, the differences in pollutant concentrations calculated by WRF-Chem were attributed to changes in the anthropogenic emissions.

## 3.3 Model validation

Several meteorological and air quality stations were used herein to evaluate the model (COVID simulation) for the lockdown (30 March to 12 April 2020) and relaxation (18 to 30 May 2020) periods. The same model configuration has been evaluated previously over the AMB for the meteorology (Ribeiro et al., 2021; Segura et al., 2021) and the chemistry, without any reduction in anthropogenic emissions (Badia et al., 2021), for different periods.

### 3.3.1 Meteorology

The meteorological data used to validate our model were from the Xarxa d'Estacions Meteorològiques Automàtiques (XEMA). Stations within this network are classified as urban and rural according to the land use of the model. Herein, we used data for the wind speed (WS), temperature (T) and relative humidity (RH). Tables S3 and S4 in the Supplement present statistical evaluations of hourly data for the Metropolitan Area of Barcelona (AMB) and Catalonia (CAT) region for the lockdown (30 March to 12 April) and relaxation (18 to 30 May) periods, respectively.

The validations of the WRF-Chem simulation (COVID run) revealed that the model generally reproduced the air temperatures of the two simulation periods well, but the performance was poorer in representing the relative humidity and the wind speed. For the first period, the simulated air temperatures for the urban stations showed low positive biases of 0.4 °C and 0.3 °C for the AMB and CAT, respectively, and an RMSE of 1.4 °C. The rural stations presented a higher bias inside the AMB (0.9 °C), which resulted from erroneous descriptions of the land use at the model resolution level (3 km). Loose performance was found for the relative humidity and the wind speed, with average RMSEs of 12.2% and 2.5 m/s, respectively. The model underestimated the relative humidity in the urban and rural areas of the AMB by 2.0% and 5.2%, respectively, while it overestimated the humidity in the rural areas of the CAT (1.4%). In the case of the wind speed, the model overestimated the wind flow over the entire domain (1.5 m/s on average in CAT), especially in the rural areas of CAT (1.9 m/s). For the second study period, a similar performance was obtained for the air temperatures inside the AMB, with a slight increase in the RMSE (1.5 °C and 1.6 °C for urban and rural areas) and a decrease in the correlations between modelled and observed data (0.90 and 0.92, respectively). Unlike the first period, the model overestimated the relative humidities at all stations in the second period, except for the rural stations in the AMB (-3.8%). The model provided lower overestimates for the wind speeds during the second period (1.0 m/s on average in CAT), although the correlations decreased in the second period.

### 3.3.2 Air quality

Air quality data from the monitoring stations Xarxa de Vigilància i Previsió de la Contaminació Atmosfèrica (XVPCA) were used here. Stations in this network were classified into different groups: urban background, urban traffic, suburban background, and rural. Here, we used the data for $O_3$ and $NO_2$. Tables S5 and S6 in the Supplement present statistical evaluations of





hourly data for the AMB and Catalonia during the lockdown (30 March to 12 April) and relaxation (18 to 30 May) periods, respectively. The modelled concentrations were converted to units of $\mu g\ m^{-3}$ by using the temperatures and pressures from the model.

Overall, the model (COVID simulation) showed reasonable agreement with the observations for $NO_2$ and $O_3$ concentrations during both periods. The best performance in the lockdown period was observed over the urban background (R between 0.43 and 0.45 for $NO_2$ and between 0.70 and 0.73 for $O_3$), while low R values were found over the rural areas (0.32 for $NO_2$ and 0.42 for $O_3$). The performance for the relaxation period was not as good as that for the lockdown period, with R values between 0.24-0.40 for $NO_2$ and 0.42-0.62 for $O_3$. However, there were negative and positive biases in both periods for $NO_2$ (NMB

between -0.15 and -0.66) and $O_3$ (NMB between 0.13 and 0.28), respectively. Similar biases were seen in another study (von Schneidemesser et al., 2021). Part of our model bias was attributed to the 1) boundary conditions used for this study (WACCM model) that added a bias to the $O_3$ background levels (Giordano et al., 2015) and 2) the current emission inventory was too coarse to accurately represent the spatial distributions and temporal variations in $NO_x$ emissions, e.g., from road transport. Low values for the modelled $NO_x$ levels underestimated ozone loss via NO titration, which resulted in high nighttime surface

ozone concentrations. The lifetime of surface $NO_x$ (few hours) is shorter than that of $O_3$ (days or weeks); thus, the surface $NO_x$ concentrations are very sensitive to emissions. We should also mention that there might be large uncertainties for the calculations of emissions factors, as discussed in Doumbia et al. (2021); underestimates of traffic $NO_x$ emissions over Europe have been mentioned previously in several air quality modelling studies (von Schneidemesser et al., 2021; Karl et al., 2017).

## 4   Results

### 4.1   Air quality changes

In the first period (30 March to 12 April), the results (difference between the COVID and BAU runs) showed a general reduction in $NO_2$ concentrations all over the Catalonia region at the surface level, with high reductions found during the evening peaks (19-21 UTC) and over the AMB (-2 to -18 $\mu g\ m^{-3}$, -10 to -70%) (see Fig. S3 in the Supplement). The highest reductions were found around the airport due to a significant reduction in air traffic emissions (see Figure 5). The surface concentrations of

VOCs were slightly lower during the morning peak (up to -2 $\mu g\ m^{-3}$, -10%) (see Fig. S4 in Supplement). During the evening peak, there were also decreases seen for several areas of the AMB and Catalonia (up to -1.5 $\mu g\ m^{-3}$, -12%). However, we also observed slight increases in some areas (up to 0.1 $\mu g\ m^{-3}$, 1%). Note that during the lockdown, the VOC emissions increased in the stationary combustion sector (see Figure 5). Changes in emissions that showed a significant decreases in $NO_2$ concentrations and slight decreases in VOC concentrations enhanced $O_3$ levels over the AMB. This is consistent with

the observations (see Figs. S1-S2 in the Supplement). The reduced $O_3$ production resulting from reductions in the levels of the $O_3$ precursors was overwhelmed by a reduction in the extent of NO titration, resulting in a net increase in $O_3$ levels. During the evening peaks (19-21 UTC), we found the highest changes (1 to 18 $\mu g\ m^{-3}$, 1 to 20%). However, when surface $O_3$ concentrations were higher (afternoon peak, 13-15 UTC), the increases were much lower (up to 6 $\mu g\ m^{-3}$, 6%) than those for the evening peak (see Fig. S5 in Supplement). Outside the AMB, the concentrations did not differ significantly (< 2 $\mu g$





m$^{-3}$, < 2%) for the two simulations. Differences in the O$_x$ (NO$_2$ + O$_3$) values were calculated to aid our interpretation of the O$_3$ concentrations by diminishing the effect of O$_3$ titration by NO in highly polluted areas (see Fig. S6 in Supplement). The overall changes in the O$_x$ concentrations remained practically constant due to a balance between the increases in O$_3$ levels and decreases in NO$_2$ levels. This has important policy implications because one air pollutant problem is being replaced by another. A similar result was seen by von Schneidemesser et al. (2021) in their study of Berlin during the lockdown.

The differences between the BAU and COVID simulations for the second period (18 to 30 May) showed overall reductions in the NO$_2$ (-2 to -15 $\mu$g $^{-3}$, -10 to -65%) and VOC levels (up to -2 $\mu$g m$^{-3}$, -16%), with high reductions found during the evening peaks (see Fig. S3 in the Supplement). Ozone levels decreased (by up to 3.5 $\mu$g m$^{-3}$, see Fig. S5 in the Supplement) in most of Catalonia due to significant reductions in most of the emission sectors (see Figure 5) during the COVID simulation, which decreased the high ozone productivity normally seen for this time of the year. However, we still found enhanced O$_3$

levels around the Barcelona airport in the evenings; the reductions in emission levels were still significant (more than 80%, see Figure 5) and inhibited titration of the O$_3$ by NO. Note that in this case, the O$_x$ concentrations decreased nearly everywhere in the Catalonia area and up to -4 $\mu$g m$^{-3}$ over the AMB (see Fig. S6 in Supplement) for the COVID simulation, resulting in overall improvements in the air quality.

## 5   Discussion

### 5.1   O$_3$ Sensitivity to precursors and land-use

Variations in the levels of the O$_3$ precursors (NO$_x$ and VOCs) had large effects on O$_3$ production. We represent this complex relationship in Figures 6 (6:00 to 8:00 UTC), 7 (13:00 to 15:00 UTC), and 8 (19:00 to 21:00 UTC), which show the differences in surface NO$_x$, VOC, and O$_3$ concentrations in the BAU and COVID simulations during the first period (30 March to 12 April, only weekdays) and the second period (18 to 30 May, only weekdays). Each dot of the top row corresponds to the O$_3$

concentration difference (ppb) of one grid cell of the AMB at the surface level. To complement this information, we calculated the average NO$_x$/VOC ratios for these two periods in Table 3. The dots in the lower row represent the land use for each grid cell, which is the key to understanding how industrial, open urban, compact urban, water, agriculture, natural open and forestland uses influenced the O$_3$ regimes (see Figure S11 and Table S9 in the Supplement for more detail on the land use classification). Values of NO$_x$ and VOC concentrations and relative changes for each land use are shown in Tables S7 and

S8 in the Supplementary Information. In Figure 6-8, we indicate the NOx-limited regime with a dark solid line separating VOC:NO$_x$ >8, which is typical for locations located downwind of urban and suburban areas, and the VOC-limited regime (VOC:NO$_x$ <8) which is typical for highly polluted urban areas (Sillman, 2003). We also indicate the transitional regime with two dotted lines (VOC:NO$_x$ >4/1 and VOC:NO$_x$ <15/1) showing where ozone becomes less sensitive to NO$_x$ changes and increases with increasing VOC levels, as identified in other studies (Wang et al., 2021a; Yang et al., 2021).

Overall, without any reduction in emissions (BAU simulation), this analysis indicates that in green areas, such as urban forests far from anthropogenic sources and influenced by high biogenic VOC emissions, ozone production is NO$_x$-sensitive in the mornings and afternoons. With the inland sea-breeze fronts seen in the late afternoons/early evenings (Massagué et al.,





2019), pollutants are transported from their sources (urban areas) to other areas (green areas). Consequently, we found a transition to a VOC-limited regime in green areas in the evenings. Some of the grids classified as naturally open and agriculture

are close to the Barcelona airport (high $NO_x$ sources), and the transport of pollutants (driven by the wind speed and direction) has a significant impact on their regime. However, we see that most of the grids classified as naturally open are NOx-limited all day. In the case of grids classified as agriculture, we see that in the morning and evening, most of these grids are in the transition or VOC-limited regimes, while in the afternoon, these grids are in the transition or NOx-sensitive regimes. Areas close to highly polluted areas (including urban areas such as compact urban, industrial, and water areas) are in VOC-sensitive

or transitional regimes all day, especially during the evening due to high traffic emissions. Note that "water points" are located around the harbour and the airport and are typically high $NO_x$ sources. In terms of ozone levels, high values are found during the morning (40-43 ppb) and evening (46-47 ppb) hours in suburban areas (forest and natural open) because there is less NO (because of less traffic) and thus less ozone degradation. In the afternoon, high $O_3$ levels are found everywhere in the AMB (49-55 ppb), especially over the urban areas (industrial, open urban and compact urban).

With the significant decrease in $NO_x$ (20-40%) and a slight decrease in VOC (1-5%) levels during the lockdown, the $O_3$ levels increased for the COVID run (1-5%), and chemical formation tends to enter the NOx-sensitive regime, especially in the morning and afternoon hours during April-March. In the case of green areas (forest, natural open and agriculture), we see clear transitions towards NOx-sensitivity regime. However, despite the cuts in emissions, most of the grids close to highly polluted areas (compact urban, industrial, and water) were still in the VOC-sensitive or transitional regimes all day, especially in the

evenings (high traffic emissions), for which we found the highest ozone increases (2.4-5%). Similar results were found when we compared both runs (BAU and COVID) for the period in May in terms of the changes in the chemical for each land use. However, we found that during May, the maximum ozone levels decreased in the COVID run during the afternoons (up to 1.6% in green areas), which was attributed to reductions in the anthropogenic emissions that decreased the ozone precursor levels (24 to 40% for $NO_x$ and 24 to 40% for VOCs) and consequently ozone production. For green areas far from anthropogenic

sources (forests), the ozone levels were also reduced in the mornings and evenings during this period.

The lockdown measures inhibited NO titration of the $O_3$, mainly due to changes in the local $NO_x$ emissions resulting from road transport. This resulted in an increase in the $O_3$ levels during the evening hours, where there was no photolytic reaction with $NO_2$, in urban areas with high population densities. We found that air quality policies based solely on transport reduction (as illustrated by the COVID lockdowns, which reduced $NO_x$ levels) actually intensified $O_3$ levels over urban areas, indicating

the need for a protocol with strident control measures to reduce $NO_x$ emissions without significantly reducing anthropogenic VOCs to control $O_3$ levels. However, high ozone production during May was reduced due to reduced levels of the precursors, and consequently, there were reductions in the maximum ozone levels for that period.

### 5.2 Impacts on the atmospheric oxidation capacity

In addition to understanding changes in the levels of $O_3$ precursors, it is important to determine how emission changes affect

the atmospheric oxidation capacity (AOC) because this plays an important role in the loss and production rates of primary and secondary pollutants. We saw increases in the concentrations of the oxidants OH and $NO_3$ during the period March-April,





mainly over the AMB, as shown in figures 9 and 10, where the left-hand panels indicate absolute concentrations (COVID-BAU) and the right-hand panels indicate relative changes in comparison to the BAU simulations ((COVID-BAU)/BAU×100). OH is the dominant tropospheric daytime oxidant, and it increases considerably (up to 0.12 ppt, +45%) because of significant reductions in $NO_2$ levels, since $NO_2$ is the primary OH sink (Elshorbany et al., 2009). The rises in these free radical levels could be the leading cause for the diurnal $O_3$ increases given their strong link with $O_3$ production (VOC and CO oxidation by OH are the initial reactions for ozone formation). In addition, the $NO_3$ radical, which is a primary night-time oxidant, also increases in areas close to the airport and harbour (4 ppt, 210%). This increase can be explained by reductions in the VOC and $NO_2$ levels, which are important sinks for $NO_3$ radicals (Elshorbany et al., 2009; Saiz-Lopez et al., 2017).

During the period in May, we also found increases in the AOC (and $O_3$), mainly for areas close to the airport and harbour, where there were still important decreases in $NO_x$ emissions. In these areas, OH levels increase up to 0.3 ppt (55%) in the afternoon, and $NO_3$ increases up to 4 ppt (230%) in the evening. However, other areas showed general decreases in AOC (-0.1 ppt for OH and -2 ppt for $NO_3$), resulting in decreases in the $O_3$ levels. Note that for both periods, the decrease in shipping emissions (a source of $NO_x$) led to increases in the levels of both radicals along the ship tracks.

Our results indicate that changes in the anthropogenic emissions lead to significant changes in the OH and $NO_3$ radical budgets, which in the case of emission reduction, such as that experienced during the COVID lockdown, lead to enhanced oxidation efficiency in the urban atmosphere of the AMB and $O_3$ enhancements. However, during the period when $O_3$ formation increased (May), there was a decrease in the AOC and $O_3$ levels (except in the airport and harbour areas) during the COVID run. The elevations of $O_3$ and AOC levels occurred because these areas were still VOC-limited regimes during this period. In terms of air quality policy, it is important to consider the budgets of free radicals so that mitigation strategies are not counterproductive.

### 5.3 Pollution transport from urban to rural

In addition to studying the mechanisms for ozone formation in the AMB, we also explored how ozone is transported to rural areas to determine the influence of urban pollution. Rural areas far from the city, such as the Vic Plain and the Pyrenees mountain range, are frequently affected by the atmospheric plume transported northwards from the AMB (Massagué et al., 2019). Indeed, ozone exceedances over these places occur when there are high levels of $NO_2$ (mainly due to road traffic) over the AMB (see section 2). The urban plume is driven inland by southeast and southern combined sea-valley-mountain breeze winds, channelled by north–south valleys, and crosses the coastal and precoastal Catalan Ranges to an intramountain plain. To understand how emission reductions in the AMB can affect the ozone levels in these rural areas, we analysed the differences in ozone concentrations between the COVID and BAU runs for four specific days during which the air masses flowed from the AMB northwards to the Pyrenees (as shown in Figures 3 and 4) and generated high levels of $O_3$ pollution in the rural areas (see Table S1 and Figures S7-S10 in the Supplement): on April 3rd and 6th there was no precipitation, with slightly high temperatures and low to moderate wind intensity, and on the 22nd and 26th of May there was warm, sunny weather and anticyclonic conditions (see section 2).





On the 3rd of April, we found significant differences between the COVID and BAU runs during the afternoon and especially in the evening, when $O_3$ differences increased up to 2-3 ppb and higher enhancements were found over the Vic Plain (from the surface up to 2000 m), as shown in Figure 11. High ground-level ozone concentrations over the Vic Plain could have been affected by vertical recirculation of the air mass with $O_3$ reservoirs from the upper layers (see Fig. S7 in the Supplement) to the lower troposphere (Querol et al., 2017; Massagué et al., 2019). Later at night, ozone accumulated on the surface following a decrease in the PBL height (PBLH), and it was removed by deposition and titration. However, the reduction in daytime $NO_x$ emissions in the COVID simulations resulted in less titration capacity and consequently an increase in the ozone levels (1-2 ppb) that remained at the surface layer along the different sites on the S–N trajectory connecting the AMB, Vic and the Pyrenees. For the 6th of April, differences between the COVID and BAU runs were only found during the evenings and later in the nights, and the ozone levels increased in the COVID run (up to 2 ppb at the surface level) from the city of Barcelona to the mountain area of Montseny (Fig. 12). On that day, this increase was not seen for the sites located farther north. Note that ozone decreases (1-2 ppb) were seen in the free troposphere, especially at night.

We found significant decreases in ozone levels in the COVID simulation (up to 3 ppb, from the surface up to 2000 m) from the AMB north to the mountain area of Montseny on both days in May (22nd and 26th), especially during the afternoons when the PBLH was the highest and solar radiation led to enhancement of the sea breeze front, which provided favourable conditions for regional transport (Massagué et al., 2019). The decreases in ozone precursor emissions (COVID simulation) resulted in less ozone production, and consequently, the ozone concentrations decreased (discussed in sections 5.1 and 5.2). This decrease was also seen for the evening hours. Note that at night, when ozone accumulated on the surface following the decrease in the PBLH, there was a slight increase in its levels due to limited titration by NO. However, there were still reductions in the ozone levels from 500 m to 2000 m at night.

Our results showed that reduced ozone precursor levels increased the ozone levels during the evenings and nights in the COVID run due to reductions in the ozone titration process during the period of April-March, not only in urban areas but also in rural areas such as Montseny, Vic Plain, and the Pyrenees. During May, the emissions reductions in the COVID simulation decreased the ozone maximum levels (in comparison with the BAU run) in the afternoon from the city of Barcelona to the mountain areas of Montseny, the Vic Plain, and the Pyrenees, thereby improving the air quality in all these areas. A comparison of these two periods, April-March and May, showed that the mitigation strategies designed to reduce the high ozone levels were more efficient in May, when ozone formation was high (high biogenic emissions coinciding with anticyclonic conditions). Thus, given the importance of meteorology in air pollution events occurring over urban and rural areas, new mitigation strategies are needed to improve the air quality and would result in significant $O_3$ reductions; the local $O_3$ coming from the AMB plume would be reduced, as would the recirculated $O_3$ and thus the intensity of surface $O_3$ fumigation from high $O_3$ reservoir layers in other areas.





## 6 Conclusions

Improving air quality is a top priority in urban areas and requires a better understanding of how the $O_3$ levels respond to changes in the emission levels of the precursors, as well as the ozone formation regimes and the atmospheric oxidation capacity and associated $O_3$ formation. Furthermore, urban emissions affect the $O_3$ levels in rural areas outside the cities. In this study, we

used the air quality model WRF-Chem to analyse the air quality changes occurring over the Metropolitan Area of Barcelona and other rural areas affected by transport of the atmospheric plume from the AMB during mobility restrictions.

The large reduction in $NO_x$ levels (up to 60%) seen during the lockdown period combined with a slight change in VOC levels (up to 10%) led to increased $O_3$ concentrations (up to 20% in the evening). The significant increase found in the evening was mainly due to reduced $O_3$ titration by NO, which prevailed over the lower $O_3$ production level caused by decreases in

the levels of the $O_3$ precursors. The lockdown occurred during April-March when ozone photochemical production was still not at the highest level. In addition, our results showed a significant increase in the atmospheric oxidation capacity (AOC) indicated by the enhanced oxidant (OH and $NO_3$) levels, which was consistent with the slight increases seen in the maximum $O_3$ concentrations during the lockdown. We also found that for several days, these increases were seen further north in rural areas such as the Vic Plain, which produced the most annual exceedances in Spain. Large enhancements over these areas

were the result of 1) a higher regional $O_3$ background level, 2) vertical recirculation of the air masses that transport high concentrations of $O_3$ from the upper levels to the lower levels, and 3) the contributions of the AMB pollution plume travelling along the S–N valley connecting the AMB and the Vic Plain and the Pyrenees. High ozone levels seriously affect human health and the environment. In addition, the consistent differences seen in $O_x$ ($NO_2 + O_3$) concentrations during the period April-March have important policy implications, i.e., that effective mitigation strategies designed to reduce air pollutants and their

health effects should include reductions in both $O_x$ and VOC levels to avoid increases in ozone levels.

During a period in May exhibiting high ozone production (high biogenic emissions and intense sunlight), decreases in the levels of the ozone precursors $NO_2$ and VOCs consequently decreased the maximum ozone (up to 4 $\mu g\ m^{-3}$) and $O_x$ (up to 4 $\mu g\ m^{-3}$) concentrations seen over Catalonia. This was consistent with the unchanged or decreased AOCs. For several days, this decrease in the ozone level was seen further north in the rural areas. In this period, we also found ozone enhancements in

the evening, mainly due to reduced $O_3$ titration by NO.

Furthermore, this analysis suggests that there was a tendency for both periods to move towards a NOx-sensitive regime. However, some areas (open urban, compact urban, industrial, water) were still under VOC-limited or transition regimes despite the remarkable $NO_x$ reduction.

We propose that measures intended to reduce ozone precursor emissions ($NO_x$ and VOC emissions) while maintaining stable

$O_3$ formation levels in the AMB would result in important reductions in $O_3$ levels in both urban and rural areas, especially in the spring-summer when the ozone productivity is the highest. The current policies based on reducing transportation-related emissions alone could unintentionally increase the AOCs in and around large cities, thereby increasing the ozone levels. We also find that air quality policies must be designed in accordance with the VOC/$NO_x$ ratio, which dictate the $O_3$ sensitivity particular to that city. Furthermore, the significant effect of NO titration demonstrates the importance of defining mitigation



strategies focused on VOC reductions. We also propose that more measurements of VOC levels are required to constrain the models representing these chemical processes, given the complexity of the relationship between these pollutants.

*Code availability.* The WRF-Chem model code is available from http://www2.mmm.ucar.edu/wrf/users/download/get_sources.html (last access: 2 June 2022), with the specific code used in this study available from the authors upon request (alba.badia@uab.cat).

*Data availability.* Specific data used in this study available from the authors upon request (alba.badia@uab.cat).

*Author contributions.* AB carried out all the model simulations and data analysis, and led the interpretation of the results and prepared the manuscript with contributions from all co-authors. GV contributed to the interpretation of the results and provided extensive comments on manuscript. RC run the Flexpart model and analysed the output. SV analise the synoptic situation of the domain. VV and RS validate the meteorology of the model with observations.

*Competing interests.* The authors declare that they have no conflict of interest.

*Acknowledgements.* This work has been made possible thanks to the financial support of the European Research Council (ERC) Consolidator project: Integrated System Analysis of Urban Vegetation and Agriculture (818002-URBAG), the Spanish Ministry of Science, Innovation and Universities, through the "Maria de Maeztu" programme for Units of Excellence (CEX2019-000940-M), and the funding and recognition awarded to the research group Sostenipra (2021 SGR 00734) by the Department of Research and Universities of the Generalitat de Catalunya. The authors thankfully acknowledge the computer resources at PICASSO and the technical support provided by the Universidad de Málaga
(RES-AECT-2020-2-0004). The authors further wish to thank XVPCA for the provision of measurement stations. Also, thanks to the free use of $HERMESv3_GR$ and the developing team for their support. We also thank the Copernicus Global and Regional emissions service for the emission inventory. All the numerical analysis were performed with the HTCondor cluster hosted by the Port d'Informació Científica (PIC). The authors also thank Qinyi Li and Xavier Querol their constructive suggestions and feedback during this study.



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



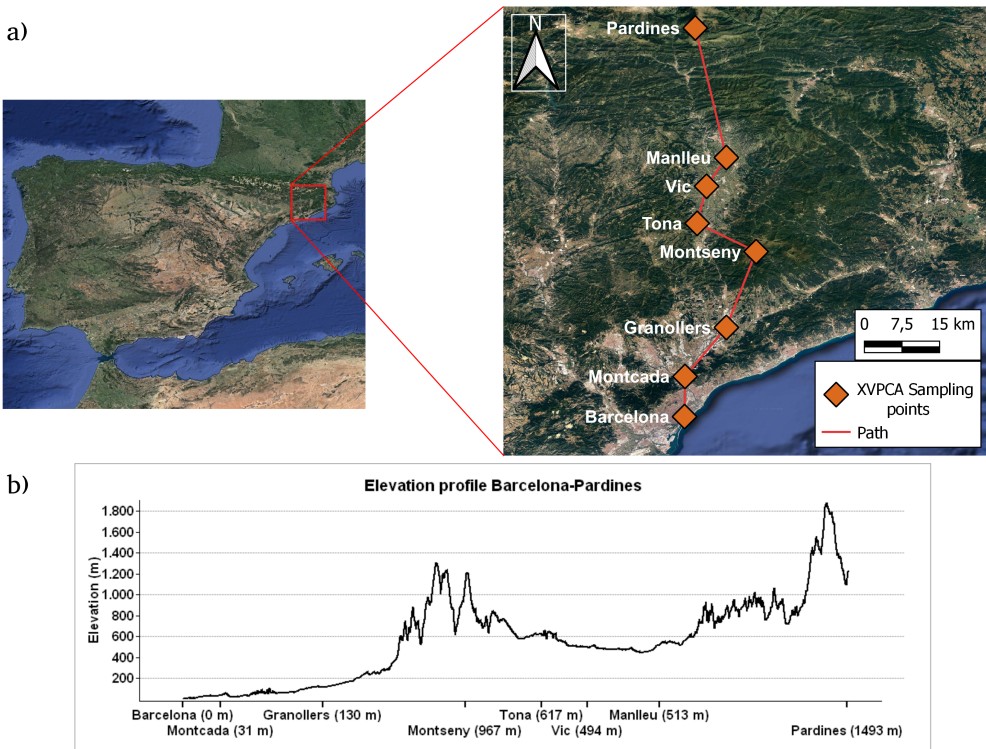

**Figure 1.** a) Location and b) main topographic features of the study area. Base maps in Panel a were taken from © Google Earth. The locations of air pollution monitoring stations (Xarxa de Vigilància i Previsió de la Contaminació Atmosfèrica, XVPCA) along the S–N axis (Barcelona-Vic Plain-Pyrenean range) are shown in Panel a (right).



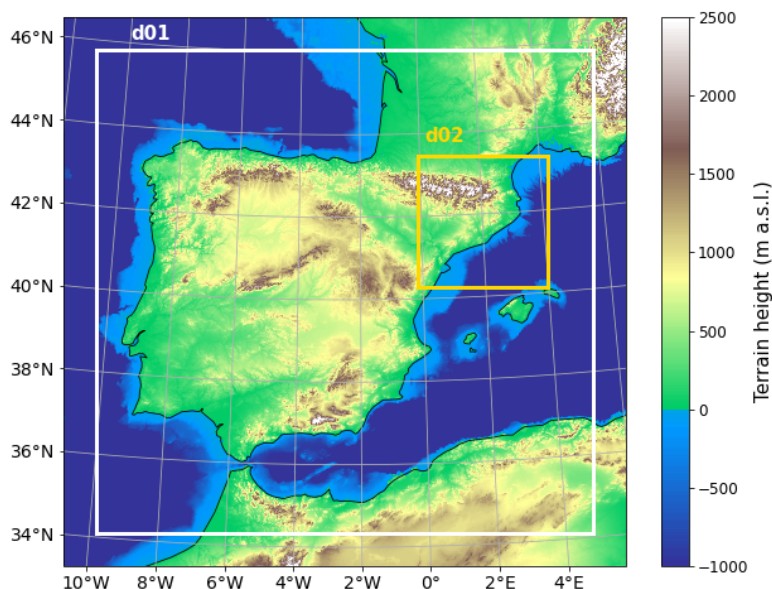

**Figure 2.** Model domains: D1: Iberian Peninsula 9 km x 9 km; D2: Catalonia horizontal resolution 3 km x 3 km.





**Figure 3.** Simulated air parcel trajectories at the footprint layer (0-300 m agl, top panels) and interlayer (300-2000 m agl, bottom panels) for days 3 and 6 of April at 16 h at the four sites (from left to right): Barcelona, Montseny, Tona (Vic plain) and Pardines.





**Figure 4.** Same as Figure 3 for days 22 and 26 of May.





**Figure 5.** Emissions reduction percentage (%) for each sector. Note that the other stationary combustion sector has a different reduction level for each pollutant. The periods analysed here are written at the bottom of the figure.



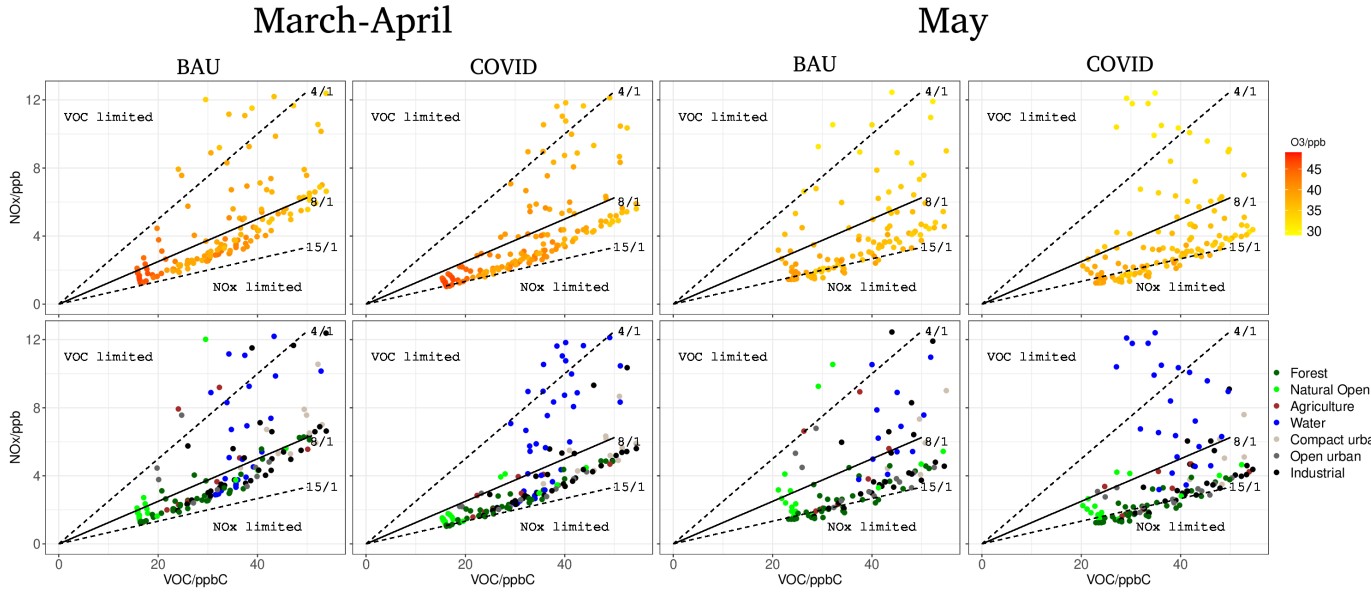

**Figure 6.** Change in O$_3$ concentrations (top panels) for 30 March to 12 April (only weekdays) and 18 to 30 May (only weekdays) for both simulations, BAU (left panels) and COVID (right panels), over the AMB area during the morning (6-8 UTC). The land use is also displayed for each grid (bottom panels).





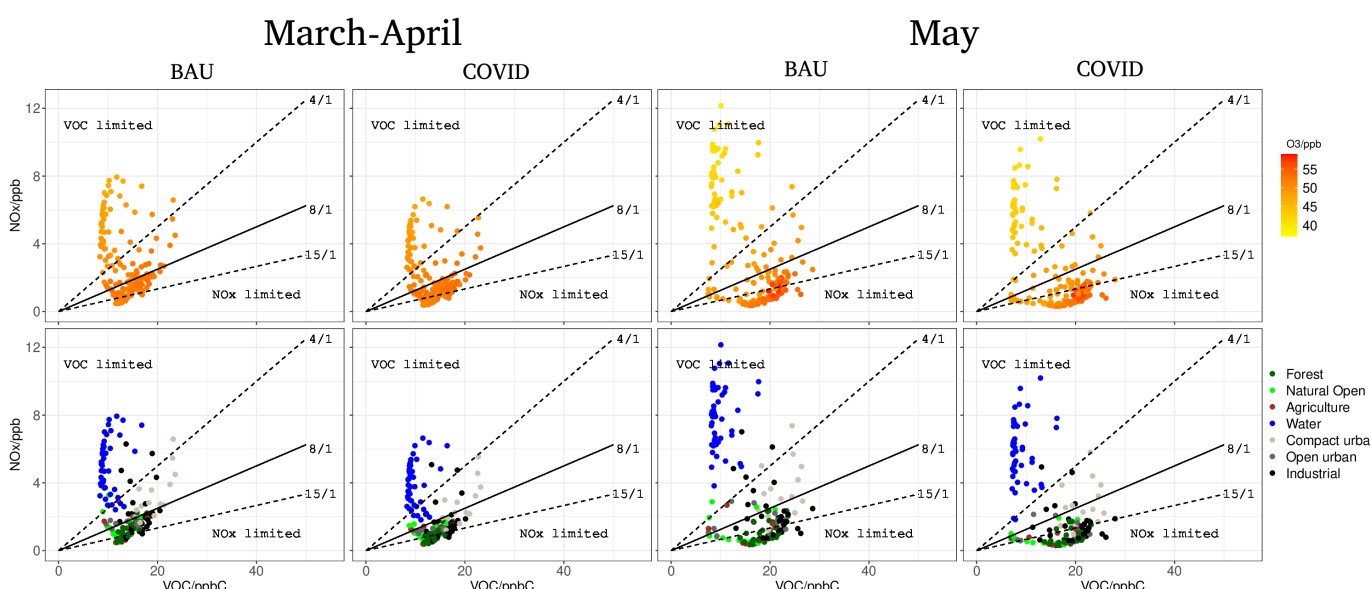

**Figure 7.** Same as Figure 6 during the afternoon (13-15 UTC).





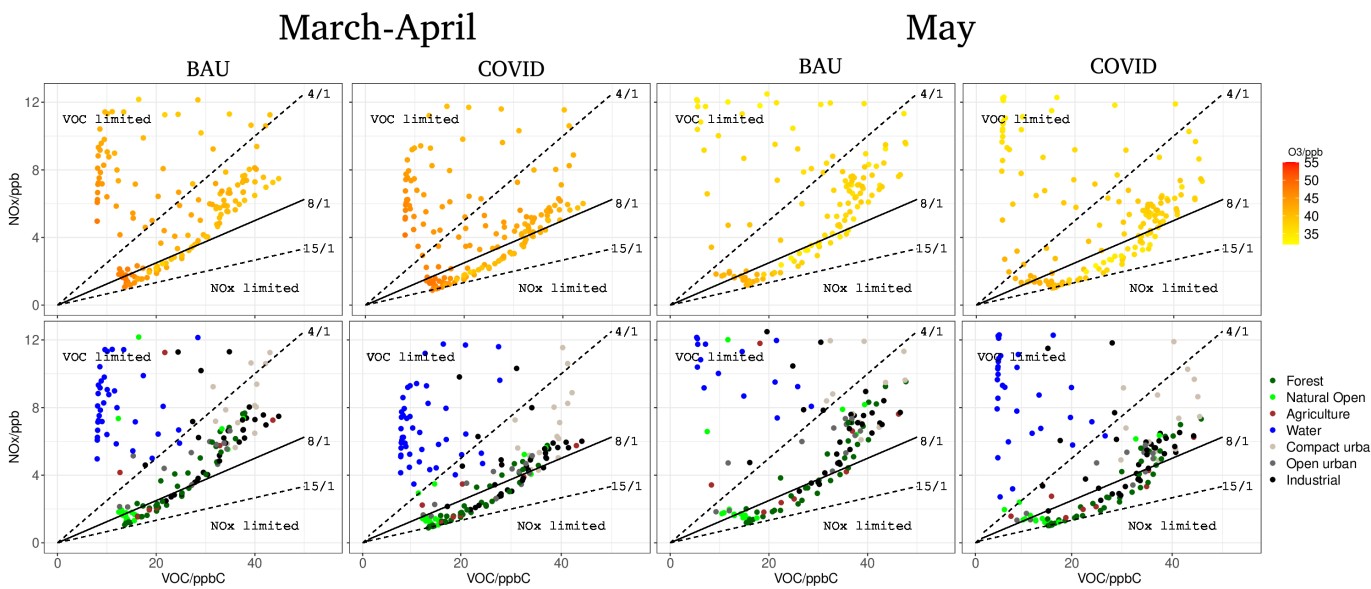

**Figure 8.** Same as Figure 6 during the evening (19-21 UTC).





# March-April

# May

**Figure 9.** Morning and afternoon averaged OH changes over the Metropolitan Area of Barcelona (AMB) and the Catalonia region during 30 March to 12 April (only weekdays) and 18 to 30 May (only weekdays), with absolute values (ppt) and relative changes (%) shown. Relative changes (%) waswere calculated as ((COVID-BAU)/BAU)×100.




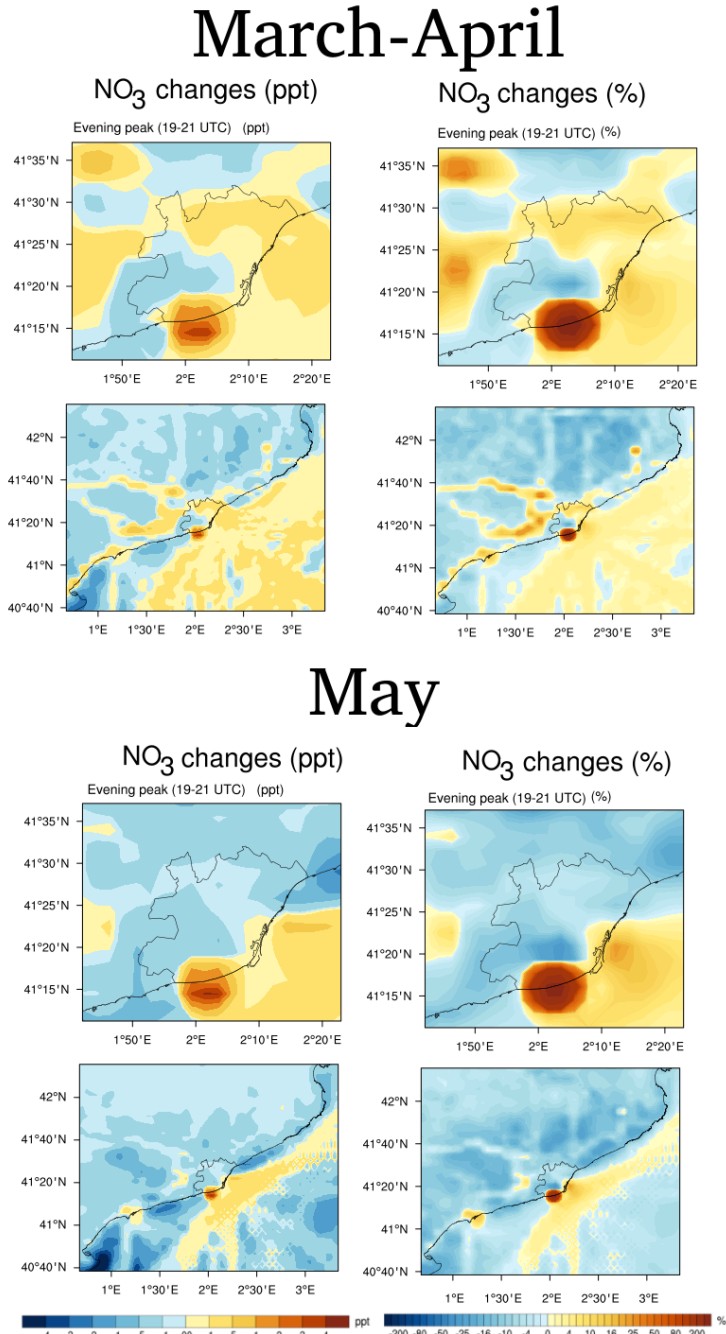

**Figure 10.** Evening-averaged NO$_3$ changes over the Metropolitan Area of Barcelona (AMB) and the Catalonia region during 30 March to 12 April (only weekdays) and 18 to 30 May (only weekdays), with absolute values (ppt) and relative changes (%) shown. Relative changes (%) were calculated as ((COVID-BAU)/BAU)×100.







**Figure 11.** O$_3$ changes between the COVID and BAU simulations along the atmospheric plume from the AMB to the Pyrenees for the 3rd of April. The modelled PBLH is shown with a green line.





**Figure 12.** Same as Figure 11 for the 6th of April.



**Figure 13.** Same as Figure 11 for the 22nd of May.





**Figure 14.** Same as Figure 11 for the 26th of May.



**Table 1.** Meteorological conditions prevailing during the COVID-19 mobility restrictions (2020). The meteorological data were extracted from the following source: Servei Meteorològic de Catalunya (SMC). The circulation weather types (CWTs) were calculated by using the surface pressures of the NCEP/NCAR reanalysis dataset. CWTs: Cyclonic 56 (C), anticyclonic (A), pure advectives (N, S, E, W, NE, SE, NW, SW), hybrid cyclone advectives (CN, CNE, CE, CSE, CS, CSW, CW, CNW), hybrid anticyclone advectives (AN, ANE, AE, ASE, AS, ASW, AW, ANW). The mobility reduction over the AMB is provided in the 4th column (source: Google mobility reports (https://www.google.com/covid19/mobility/)).

| Week | Dates (2020) | Stage | Mobility reduction | Psfc (hPa) | T (ºC) | Precipitation accumulated (mm) | CWTs |
|---|---|---|---|---|---|---|---|
| 1 | 16-22/03 | Lockdown | 75.4% | 1020,4 | 12,9 | 24,1 | SE |
| 2 | 23-29/03 | Lockdown | 82.2% | 1015,8 | 11,4 | 20,4 | E |
| 3 | 30/03-05/04 | Full Lockdown | 84.4% | 1015,9 | 11,6 | 25,3 | C |
| 4 | 06-12/04 | Full Lockdown | 84.4% | 1023,0 | 15,0 | 0 | A |
| 5 | 13-19/04 | Lockdown | 82.6% | 1014,5 | 15,1 | 109,1 | U |
| 6 | 20-26/04 | Lockdown | 79,8 % | 1011,1 | 15,7 | 126,7 | C |
| 7 | 27/04-03/05 | Lockdown | 74% | 1013,8 | 18,5 | 2,3 | N |
| 8 | 04-10/05 | F0 | 60% | 1014,0 | 19,3 | 6,9 | U |
| 9 | 11-17/05 | F0 | 58,2% | 1010,0 | 17,6 | 13,7 | C |
| 10 | 18-24/05 | F0-F1 | 50,6% | 1020,4 | 21,8 | 0,0 | U |
| 11 | 25-30/05 | F1 | 43,2% | 1021,0 | 20,8 | 4,6 | E |
| 12 | 01-07/06 | F1-F2 | 41,4% | 1009,5 | 19,8 | 21,9 | U |
| 13 | 08-14/06 | F2 | 39% | 1012,0 | 19,2 | 24,5 | C |
| 14 | 15-20/06 | F2-F3 | 34,8% | 1018,1 | 20,5 | 16,4 | U |



**Table 2.** Model details and experiment configuration

|  |  |
|---|---|
| **Chemistry** | |
| Chemical mechanism | RADM2 (Stockwell et al., 1990) |
| Aerosol scheme | MADE/SORGAM aerosol scheme (Ackermann et al., 1998; Schell et al., 2001) |
| Photolysis scheme | Fast-J (Wild et al., 2000) |
| Dry deposition | Wesely (2007) |
| Wet deposition | Grell and Dévényi (2002) |
| Anthropogenic emissions | CAMS-REG-APv3.1 database (Granier et al., 2019) |
| Biogenic emissions | MEGAN (Guenther et al., 2012) |
| **Physics** | |
| Urban canopy scheme | BEP+BEM (Salamanca et al., 2011) |
| PBL scheme | BouLac (Bougeault and Lacarrere, 1989) |
| **Resolution and Initial conditions** | |
| Horizontal resolution | D1: 9 km×9 km, D2: 3km x 3km |
| Vertical layers | 45 |
| Top of the atmosphere | 100 hPa |
| Chemical initial condition | WACCM (Gettelman et al., 2019) |
| Meteorological initial condition | ERA5 (Hersbach et al., 2020) |
| Chemistry spin-up | 1 month |



**Table 3.** Averages NO$_x$/VOC ratio and ozone concentrations from 30 March to 12 April (only weekdays) and 18 to 30 May (only weekdays) in the morning (6-8 UTC), afternoon (13-15 UTC) and evening (19-21 UTC). Red and blue cells indicate VOCs and NO$_x$ regimes, respectively. The relative changes in ozone concentrations (%) are shown in brackets and were calculated as ((COVID-BAU)/BAU)×100.

| March-April | Landuse | Morning | | Afternoon | | Evening | |
|---|---|---|---|---|---|---|---|
| | | BAU | COVID | BAU | COVID | BAU | COVID |
| [NO$_x$/VOC] | Forest | 0.102 | 0.083 | 0.073 | 0.059 | 0.143 | 0.111 |
| | Natural Open | 0.158 | 0.098 | 0.109 | 0.077 | 0.194 | 0.122 |
| | Agriculture | 0.156 | 0.100 | 0.113 | 0.077 | 0.195 | 0.101 |
| | Water | 0.25 | 0.201 | 0.495 | 0.393 | 0.707 | 0.535 |
| | Compact urban | 0.166 | 0.139 | 0.171 | 0.137 | 0.236 | 0.184 |
| | Open urban | 0.125 | 0.088 | 0.108 | 0.081 | 0.182 | 0.135 |
| | Industrial | 0.149 | 0.120 | 0.135 | 0.100 | 0.215 | 0.151 |
| O$_3$ (ppb) | Forest | 40.3 | 40.7 (1.0 %) | 51.7 | 51.7 (0.0 %) | 42.3 | 42.9 (1.3 %) |
| | Natural Open | 42.8 | 43.9 (2.5 %) | 51.7 | 51.9 (0.3 %) | 45.4 | 46.4 (2.1 %) |
| | Agriculture | 37.3 | 38.5 (3.4 %) | 51 | 51.2 (0.4%) | 40.7 | 42.2 (3.7 %) |
| | Water | 37 | 38.7 (4.6 %) | 48.6 | 49.4 (1.6 %) | 42.1 | 44.2 (5.0 %) |
| | Compact urban | 35.8 | 36.9 (2.9 %) | 52 | 52.3 (0.6 %) | 40.4 | 42.2 (4.3 %) |
| | Open urban | 39.4 | 40.4 (2.4 %) | 52.6 | 52.8 (0.3 %) | 42.6 | 43.6 (2.4 %) |
| | Industrial | 36.8 | 37.7 (2.3 %) | 52.2 | 52.5 (0.5 %) | 40.1 | 41.8 (4.2 %) |

| May | Landuse | Morning | | Afternoon | | Evening | |
|---|---|---|---|---|---|---|---|
| | | BAU | COVID | BAU | COVID | BAU | COVID |
| [NO$_x$/VOC] | Forest | 0.076 | 0.066 | 0.053 | 0.043 | 0.139 | 0.115 |
| | Natural Open | 0.122 | 0.087 | 0.078 | 0.052 | 0.194 | 0.129 |
| | Agriculture | 0.122 | 0.090 | 0.077 | 0.046 | 0.192 | 0.118 |
| | Water | 0.349 | 0.279 | 0.792 | 0.663 | 1.327 | 1.028 |
| | Compact urban | 0.137 | 0.121 | 0.156 | 0.125 | 0.261 | 0.210 |
| | Open urban | 0.101 | 0.078 | 0.099 | 0.072 | 0.191 | 0.150 |
| | Industrial | 0.125 | 0.104 | 0.107 | 0.075 | 0.230 | 0.165 |
| O$_3$ (ppb) | Forest | 36.9 | 36.7 (-0.6 %) | 51.3 | 50.5 (-1.6 %) | 37.7 | 37.6 (-0.1 %) |
| | Natural Open | 37.0 | 37.3 (0.6 %) | 48.9 | 48.3 (-1.3 %) | 40.3 | 40.6 (0.8 %) |
| | Agriculture | 33.3 | 33.8 (1.4 %) | 50.5 | 49.9 (-1.2 %) | 36.3 | 37.3 (3.0 %) |
| | Water | 29.7 | 31.2 (4.9 %) | 42.5 | 43.4 (2.2 %) | 32.0 | 35 (9.4 %) |
| | Compact urban | 33.6 | 33.9 (1.0 %) | 50.8 | 50.4 (-0.8 %) | 35.8 | 37.2 (3.8 %) |
| | Open urban | 36.7 | 36.9 (0.5 %) | 51.3 | 50.7 (-1.1 %) | 38.2 | 38.6 (1.1 %) |
| | Industrial | 34.3 | 34.6 (0.9 %) | 52.1 | 51.6 (-1.1 %) | 36.3 | 37.6 (3.5 %) |