# Peer review of "Modelling the impacts of emission changes on $O_3$ sensitivity, atmospheric oxidation capacity and pollution transport over the Catalonia region"

_EGUsphere, 2023_

## Author Comment (AC1)

**Response to Reviewers**

We thank the reviewers for the constructive comments and suggestions which have helped us improve the manuscript. Below we give full detailed answers to each issue raised by each reviewer. Our response is in blue, to differentiate from the comment which is in black. Furthermore, we include any new text added in the manuscript in red, to facilitate this second revision.

To summarize the main changes to the manuscript,  we would like to point out:

1. We have expanded the discussion section to include 1) a more detailed description of the two different photochemical regimes that describe the sensitivity of O3 to its precursors, 2) a new figure entitled "O3 concentration as a function of VOC/NOx concentration.", 3) more description of the trajectory experiments using the FLEXPART-WRF model and 4) further justification of the chemical scheme chosen for the simulations.
2. We have rewritten the main text to clarified that changes in the oxidation capacity are related to O3 concentrations given that VOC and CO oxidation by OH are the initial reactions for ozone formation and we have expanded the discussion of the oxidation capacity.
3. We have added more references to support the main text when introducing the tropospheric ozone and O3 photochemical regimes.

**Response to comments of Reviewer 1**

This is a very nice analysis that provides a lot of useful information and insight regarding the production of ozone associated with the reduction of anthropogenic emissions during the COVID-19 pandemic, as well as the changes in the chemical regime associated with it.

**Response:** We thank the reviewer for his/her comments. Below are our point-by-point replies to each specific comment raised by Reviewer 1.

Specific comments

Line 22: Add more recent references as Fleming et al (2018), Sillman et al (2021)

Response: Thank you for suggesting more references. These two references have been added to the updated manuscript.

This section now reads: Tropospheric ozone (O3) is a radiatively active gas that acts as an oxidizing agent and a surface pollutant in urban areas, where it is a major component of photochemical smog and causes a number of respiratory health effects (Sillman, 2003; Anenberg et al., 2010; Fleming et al., 2018; Sillmann et al., 2021).

Line 70: remove 70

Response: We have corrected this typo.

Section 2: some of the discussion belongs to Introduction.

Response: Thank you for this comment. We have moved some of the discussion from Section 2 to Section 5.3 as suggested by Reviewer 2 (see next comment). We think that the rest of the text belongs to this section because describes our case study.

Lines 174-185: the discussion could be part of the supplementary material.

Response: Thank you for this suggestion. We have moved some of the discussion from Section 2 to Section 5.3 as suggested by Reviewer 2 which we found more appropriate than the Supplementary material as you suggest because that is where the discussion of the trajectory experiments is described.

The revised Section 5.3, now reads:

Figures 9 and 10 show the trajectories of the air masses arriving at the monitoring stations on the selected days, which were modelled with the Lagrangian particle dispersion model FLEXPART-WRF (Brioude et al., 2013). This version of the Lagrangian model works with the WRF mesoscale meteorological model, with the same parametrization as the WRF-Chem model (see section 3.1). The transport model has been run in backwards mode, which means that what is represented in each plot is the residence time, at each grid cell of the map, for the air masses arriving at each site. Twenty-four-hour back trajectories were calculated for each day at a release time of 16 h and with a grid cell size of 0.03 x 0.03 degrees. Figures 9 and 10, show that the air masses on the 3rd of April and 22 of May were transported from the AMB to rural areas such Montseny and the Vic Plain, and we can see an influence from the bottom layers (0-300 m) and the upper layers (300-2000 m) at the different sites. The air masses on the 6th of April were channelled from the AMB northwards to Montseny, the Vic Plain and the Pyrenees. The air masses on the 26th of May were also transported from the AMB northwards to Montseny, the Vic Plain and the Pyrenees, but the air masses that arrived at the surfaces of these locations had strong local components and larger influences from the upper layers.

Line 193: Fig 2 is refereed first time after Figs. 3 and 4

Response: Amended. We have changed the number of Figures in the updated manuscript: Figure 2 is now Figure 4 and Figures 3 and 4 are Figures 2 and 3, respectively.

Section 3.1: Mar et al (2016), Im et al (2016) showed that RADM2 underestimates the O3 concentration when compared to other chemical mechanisms. A discussion about the choice of chemical mechanism would be beneficial since it looks like the Authors obtained the right answers for the wrong reasons.

Response: Thank you for pointing this out. The chemical mechanism RADM2 has been successfully used in several studies of air quality in Europe (Im et al., 2015; Tuccella et al., 2011, Badia et al., 2021). In particular, the RADM2 chemical mechanism has been used in Badia et al., 2021 over the Metropolitan Area of Barcelona.

From Mar et at., (2016):

- *Model biases for O3 in both the MOZART and RADM2 simulations are in line with biases found in other regional modeling studies for Europe.*

- *The temporal correlation with hourly measurements for O3 in this study are also in line with other regional modeling studies of O3 for Europe.*

From Im et al (2016):

- *All models capture, reasonably well, the shape of the domain-averaged annual diurnal cycle of O3 over both domains, while the sub-regional temporal variability are simulated from moderate to good depending on the season and the sub-region that the particular model is configured for.*

Having said that, we have expanded the description of the choice of the chemical mechanism.

Section 3.1, lines 195-198: The chemical mechanism RADM2 has been broadly used in modeling studies of the air quality over Europe  (Im et al., 2015; Tuccella et al., 2011, Badia et al., 2021) and its model biases for NO2 and O3 are inline with other air quality modelling studies over Europe (Im et al., 2015,  Mar et at., 2016). In particular, the RADM2 chemical mechanism has been used in Badia et al., 2021 over the AMB.

Section 3.3: Please check the numbers in the Tables, not always the MB=MM-OM

Response: Thank you for pointing this. The numbers have been checked and updated in the manuscript.

Lines 301-314: A lot of this information should go to the Figures caption (e.g. "The dots in the lower row represent the land use for each grid cell, which is the key to understanding how industrial, open urban, compact urban, water, agriculture, natural open and forestland uses influenced the O3 regimes")

Response: Thank you for this comment. We have rewritten this part and moved information to the Figures caption.

Section 5.1, lines 301-303: In addition, the land use is the key to understanding how industrial, open urban, compact urban, water, agriculture, natural open and forest land uses influenced the $O_3$ regimes (see Figure S11 and Table S9 in the Supplement for more detail on the land use classification).

Figure 4 caption: Modelled O3 concentrations (top panels) for 30 March to 12 April (only weekdays) and 18 to 30 May (only weekdays) for both simulations, BAU (left panels) and COVID (right panels), over the AMB area during the morning (6-8 UTC). Each dot of the top row corresponds to the O3 concentration difference (ppb) of one grid cell of the AMB at the surface level. The dots in the lower row represent the land use for each model grid cell.

Line 315: please specify the land-use categories that belong to "green areas".

Response: Thank you for pointing this out. Green areas (forest, natural open and agriculture) are described later in the text (line XX ). However, we have rewritten the text to clarify that in line 315 we are talking about "urban forest":

Section 5.1, lines 318-320: Overall, without any reduction in emissions (BAU simulation), this analysis indicates that in urban forests far from anthropogenic sources and influenced by high biogenic VOC emissions, the photochemical regime of O3 formation is NOx-sensitive in the mornings and afternoons.

Section 5.1, lines 321-322: Consequently, we found a transition to a VOC-limited regime in green areas (forest, natural open and agriculture) in the evenings.

Figures 3-4: Increase the size of the cross and explain what it represents.

Response: We have increased the size of the cross and the add more information into Figures 3-4 caption.

Figure 3-4 caption: Simulated air parcel trajectories at the footprint layer (0-300 m agl, top panels) and interlayer (300-2000 m agl, bottom panels) for days 3 and 6 of April at 16 h at the four sites (from left to right): Barcelona, Montseny, Tona (Vic plain) and Pardines. The location of each site is shown with a green cross.

In the updated manuscript, Figures 3-4 are:

[Figure]

Residence time ($\log_{10}$ seconds)

[Figure]

Figure 5 Sectors A and G, B and H, as well as the pollutants CO and NOx and NH3 and PM10 have similar colors and it is difficult to distinguish between different lines.

Response: Amended. We have changed the colors and in the updated manuscript Figure 5 is:

[Figure]

[Figure]

Figures 6-8 As before, we can't really distinguish the colors. I would suggest using a discrete color scale.

Response: We use a discrete color to display the land-use for each grid (bottom panels). However, we think the ozone concentrations can not be represented in discrete color.

Figures 12-14 There is no reference to these Figures in the text.

Response: Amended. These figures are referenced in the text in Section 5.3.

Table 1 define F0, F1, F2, F3

Response: Amended. The explanation for the acronyms F0, F1, F2 and F3 have been added in Table 1 caption.
Table 1 caption: "F0, F1, F2, F3 are the different phases of the de-escalation period being F0 the first phase after lockdown and F3 being the last phase before all restrictions were eliminated".

Fleming, Z., Doherty, R., Von Schneidemesser, E., Malley, C., Cooper, O., Pinto, J., Colette, A., Xu, X., Simpson, D., Schultz, M., Lefohn, A., Hamad, S., Moolla, R., Solberg, S., and Feng, Z.: Tropospheric Ozone Assessment Report: Present-day ozone distribution and trends relevant to human health, Elementa, 6, 12, https://doi.org/10.1525/elementa.273, 2018

Sillmann, J., Aunan, K., Emberson, L., Büker, P., Van Oort, B., O'Neill, C., Otero, N., Pandey, D., and Brisebois, A.: Combined impacts of climate and air pollution on human health and agricultural productivity, Environ. Res. Lett., 16, 093004, https://doi.org/10.1088/1748-9326/ac1df8, 2021.

Mar, K. A., Ojha, N., Pozzer, A., and Butler, T. M.: Ozone air quality simulations with WRF-Chem (v3.5.1) over Europe: model evaluation and chemical mechanism comparison, Geosci. Model Dev., 9, 3699–3728, https://doi.org/10.5194/gmd-9-3699-2016, 2016.

Im, U., Bianconi, R., Solazzo, E., Kioutsioukis, I., Badia, A., Balzarini, A., Baro, R., Bellasio, R., Brunner, D., Chemel, C., Curci, G., Flemming, J., Forkel, R., Giordano, L., Jimenez-Guerrero, P., Hirtl, M., Hodzic, A., Honzak, L., Jorba, O., Knote, C., Kuenen, J.J.P., Makar, P.A., Manders-Groot, A., Neal, L., Perez, J.L., Pirovano, G., Pouliot, G., San Jose, R., Savage, N., Schroder, W., Sokhi, R.S., Syrakov, D., Torian, A., Tuccella, P., Werhahn, K., Wolke, R., Yahya, K., Zabkar, R., Zhang, Y., Zhang, J., Hogrefe, C., Galmarini, S., 2015. Evaluation of operational online-coupled regional air quality models over Europe and North America in the context of AQMEII phase 2. Part I: Ozone. Atmos. Environ. 115, 404e420.

---

## Author Comment (AC2)

**Response to Reviewers**

We thank the reviewers for the constructive comments and suggestions which have helped us improve the manuscript. Below we give full detailed answers to each issue raised by each reviewer. Our response is in blue, to differentiate from the comment which is in black. Furthermore, we include any new text added in the manuscript in red, to facilitate this second revision.

To summarize the main changes to the manuscript, we would like to point out:

1. We have expanded the discussion section to include 1) a more detailed description of the two different photochemical regimes that describe the sensitivity of O3 to its precursors, 2) a new figure entitled "O3 concentration as a function of VOC/NOx concentration.", 3) more description of the trajectory experiments using the FLEXPART-WRF model and 4) further justification of the chemical scheme chosen for the simulations.
2. We have rewritten the main text to clarified that changes in the oxidation capacity are related to O3 concentrations given that VOC and CO oxidation by OH are the initial reactions for ozone formation and we have expanded the discussion of the oxidation capacity.
3. We have added more references to support the main text when introducing the tropospheric ozone and O3 photochemical regimes.

**Response to comments of Reviewer 2**

This is an interesting paper looking at the impact of emissions reductions on atmospheric chemistry. It takes the area in and around the Barcelona metropolitan area as a natural laboratory, and studies two periods in 2020 as exemplar systems to understand the effect of emissions reductions on ozone and NO2.

The paper describes a model study using WRF-Chem coupled to an urban canopy model to look at atmospheric processes over the AMB region, and FLEXPART-WRF to do some trajectory analysis to study the chemistry occurring as air flows inland.

This is an ambitious study which aims to use the connection between the natural experiment of emissions reductions in the months of April/May 2020, and through analysis of idealised counterfactual experiments perform attribution of the effect of emissions reductions.

The experimental design and analysis appear sound and this manuscript fits well within the scope of ACP. I feel the structure of the manuscript could be improved, and the discussion should be improved in places before publication.

**Response:** We thank the reviewer for his/her comments. Below are our point-by-point replies to each specific comment raised by reviewer 2.

**Specific comments**

The title is the first area to address - I felt it was perhaps a bit too general, as the main focus is the impact of lockdown.

Response: Thank you for your comment. We feel that the main focus of this manuscript is to discuss the impact of emission reductions on ozone chemistry (O3 sensitivity, atmospheric oxidation capacity, and pollution transport from the city to rural areas). The lockdown period provided an excellent opportunity to do this because of the drastic emission reduction imposed on the city due to transport and industrial activity restrictions. It gave us an unprecedented opportunity to study these mechanisms, but our focus is not on the air quality consequences of the lockdown and thus feel justified in our choice of title.

The abstract can also be a bit more explicit eg L8/9 'response of ozone chemistry to changes' could make more explicit what reduction is under discussion.

Response: Thank you for this comment. We have now expanded these lines and this part of the abstract now reads:

The aim is to investigate the response of ozone chemistry to reduction of precursos emissions (NOx, VOC).

AOC needs to be defined in the abstract, and for the sake of clarity that it excludes O3 oxidant.

Response: Thank you for this comment. We have clarified that we are talking about OH and NO3 radicals here.

This part of the abstract now reads:

3) the increase in the maximum O3 level (up to 6%) during the emission-reduction period could be attributed to an enhancement in the atmospheric oxidants hydroxyl and nitrate radical (OH and NO3) given their strong link with O3 loss/production chemistry

Abstract Conclusion # 3 is not clear - what is the mechanism? Conclusion #4 could the authors explain why May is important? Conclusion 5 - not sure what is meant by a change contributing to a plume.  Perhaps re-word?

Response: Thank you for these suggestions. We have rewritten the abstract to clarify these conclusions. This part of the abstract now reads:

3) the increase in the maximum O3 level (up to 6%) during the emission-reduction period could be attributed to an enhancement in the atmospheric oxidants hydroxyl and nitrate

radical (OH and NO3) given their strong link with O3 loss/production chemistry, 4) the daily maximum levels of ozone and odd oxygen species (Ox) generally decreased (4%) in May -a period with intense radiation which favors ozone production- with the reduced atmospheric OH and NO3 oxidants, indicating an improvement in the air quality, 5) ozone concentration changes in the urban plume of Barcelona contribute significantly to the level of pollution along the 150km south-to-north valley to the Pyrenees.

S2 describes the region selected for study, geography, Barcelona's air quality with respect to guidelines, the Vic Plain and the ozone situation in 20202. Two periods are identified for closer study, and also days of even closer study. I feel the manuscript would be more readable if it would it be possible to decide on a single consistent nomenclature for the two periods, eg P1 and P2 and so avoid changing between Mar-Apr/lockdown/first period and May/relaxation through the text

Response: Thank you for this comment. We feel that nomenclature of P1 and P2 is not intuitive to understand the two periods analysed, and writing out "March-April" and "May" does not occupy much more space, so we would rather cut down the acronyms and write out the periods.

L174-185 Some of this section could be grouped with the discussion of the trajectory experiments, as it mixes model description with some analysis that probably belongs with the discussion in S5.3. It would be interesting to better justify why these days were chosen for further study - what aspects do these days/analysis bring out?

Response: Thank you for this comment. We have now included more information about the selection of these days in section 2 and moved information to the discussion of the trajectory in Section 5.3. In addition, Figures 3-4 have been moved to section 5.3 and all Figures have been renumbered according to these changes.

Section 2, lines 168-172: In addition, we select two days in the lockdown period (the 3rd and 6th of April) and two days in the relaxation period (the 22nd and 26th of May), during which high ozone concentrations were registered (see Table S1 in the Supplement) and there is a clear influence of the air masses from the AMB to rural areas far from the city (discussed in section 5.3), to study the changes in the O3 circulation from Barcelona (Ciutadella) to the Pyrenees mountains (Pardines), including the Vic plane (Tona) and Montseny.

Section 5.3, lines 391-402: Figures 9 and 10 show the trajectories of the air masses arriving at the monitoring stations on the selected days; which were modelled with the Lagrangian particle dispersion model FLEXPART-WRF (Brioude et al., 2013). This version of the Lagrangian model works with the WRF mesoscale meteorological model, with the same parametrization as the WRF-Chem model (see section 3.1). The transport model has been

run in backwards mode, which means that what is represented in each plot is the residence time, at each grid cell of the map, for the air masses arriving at each site. Twenty-four-hour back trajectories were calculated for each day at a release time of 16 h and with a grid cell size of 0.03 x 0.03 degrees. Figures 9 and 10, show that the air masses on the 3rd of April and 22 of May were transported from the AMB to rural areas such Montseny and the Vic Plain, and we can see an influence from the bottom layers (0-300 m) and the upper layers (300-2000 m) at the different sites. The air masses on the 6th of April were channelled from the AMB northwards to Montseny, the Vic Plain and the Pyrenees. The air masses on the 26th of May were also transported from the AMB northwards to Montseny, the Vic Plain and the Pyrenees, but the air masses that arrived at the surfaces of these locations had strong local components and larger influences from the upper layers.

S3 describes the WRF-Chem experiments performs some model evaluation against observations. The model is shown to be more skillful in meteorology than chemistry, with ozone biases around 20-30% shown. The authors do not discuss if the bias in the model means that the model correctly simulates the difference in ozone/NO2 from a change in emissions. Given the chemistry is non-linear, would the response be greater/smaller in a less biased model? Would it make sense to compare ozone changes in the model with differences in climatology/COVID period at the observation stations of interest to assess if the model gets delta_O3 correct?

Response: Thank you for this constructive comment. As you already point out, the chemistry is non-linear and it´s not a straight forward answer. Reductions in observation stations in the city of Barcelona for NO2 and O3 concentrations in the lockdown and relaxation periods are given by Querol et al., 2021. This study calculates a reduction in NO2, with meteorology correction, between 46-50% and 19-23% for the lockdown and full relaxation periods, respectively. In the case of 8hDM O3, the changes in concentrations, with meteorology correction, are between -1-6% and -9- -2% for the lockdown and full relaxation periods. These changes are in line with our model changes for these two species (excluding the airport area that has higher changes). We have added more discussion about the model bias in the manuscript as well as added the reference of Querol et al., 2021.

Section 3, lines 263-265: Although the model exhibits these biases, the modelled air quality changes presented in the next section are in line with other studies such as Querol et al.,2021 that present a comparison between data from years 2015-2019 and the lockdown and relaxation periods for the year 2020 in the city of Barcelona.

Section 6, lines 473-475:  In addition, the difficulty of models in simulating urban ozone precursors such as NOx and VOC levels and, consequently, its link with the ozone chemistry should be addressed in future work to use the models as effective tools for assessing future studies aimed at reducing air quality.

Querol, X., Massagué, J., Alastuey, A., Moreno, T., Gangoiti, G., Mantilla, E., Duéguez, J. J., Escudero, M., Monfort, E., Pérez García-Pando, C., Petetin, H., Jorba, O., Vázquez, V., de la Rosa, J., Campos, A., Muñóz, M., Monge, S., Hervás, M., Javato, R., & Cornide, M. J. (2021). Lessons from the COVID-19 air pollution decrease in Spain: Now what? In Science of The Total Environment (Vol. 779, p. 146380). Elsevier BV. https://doi.org/10.1016/j.scitotenv.2021.146380

S4 describes the results of the experiments. This section in the MS has the most potential for improvement, I feel. Firstly, the changes in O3, NO2 and Ox are given in the supplementary. Is it worth moving figures S3 and S4 into the main text? I appreciate the changes are given in absolute (LH) and relative (RH) terms for both period 1 adn period 2. Could the labelling of these figures be improved to indicate what data are plotted in each of the four rows are? There are no labels on each of the second and fourth rows. The text in S4.1 could do with a further polish, e.g.

Response: Thank you for this comment. We feel the manuscript is already overloaded with figures (currently 14 figures), and would rather keep these two figures in supporting information. Figures S3-S6 have been improved to indicate what data is plotted in each row (AMB and CAT). We have also edited the captions to indicate that the Catalonia region is show as CAT.  The newly revised figures are:

[Figure]

Figure S3: Averaged surface NO2 changes over the Metropolitan Area of Barcelona (AMB) and the Catalonia region (CAT) during 30 March to 12 April (only weekdays) and 18 to 30 May (only weekdays) in absolute value (ug/m3) and relative change (%).  Relative change (%) is calculated as (COVID-BAU)/BAU×100..

**March-April**

[Figure]

**May**

Figure S4. Same as Figure S3 for VOC changes.

[Figure]

Figure S5. Same as Figure S3 for O3 changes.

**March-April**

[Figure]

Figure S6. Same as Figure S3 for Ox changes.

Additionally, we have applied the same changes to Figures 7 and 8, which we have been updated to indicate what data is plotted in each row (AMB and CAT) and the captions have been edited to indicate that the Catalonia region is show as CAT.

**March-April**

[Figure]

**May**

Figure 7. Morning and afternoon averaged surface OH changes over the Metropolitan Area of Barcelona (AMB) and the Catalonia region (CAT) during 30 March to 12 April (only weekdays) and 18 to 30 May (only weekdays), with absolute values (ppt) and relative changes (%) shown. Relative changes (%) were calculated as ((COVID-BAU)/BAU)×100.

**March-April**

**NO$_3$ changes (ppt)**

**NO$_3$ changes (%)**

**May**

**NO$_3$ changes (ppt)**

**NO$_3$ changes (%)**

Figure 8. Evening-averaged surface NO3 changes over the Metropolitan Area of Barcelona (AMB) and the Catalonia region (CAT) during 30 March to 12 April (only weekdays) and 18 to 30 May (only weekdays), with absolute values (ppt) and relative changes (%) shown. Relative changes (%) were calculated as ((COVID-BAU)/BAU)×100.

The text in section 4.1, has been improved:

Section 4.1: During the first period (30 March to 12 April) we see a general reduction in NO2 concentrations of the COVID simulation with respect to the BAU all over the Catalonia region at the surface level, with high reductions found during the evening peaks (19-21 UTC) and over the AMB (-2 to -18 µg m−3, -10 to -70%) (see Fig. S3 in the Supplement). The highest reductions were found around the airport due to a reduction in air traffic emissions (see Figure 3). The surface concentrations of VOCs were slightly lower during the morning peak, with reductions up to -2 µg m−3 (-10%) as can be appreciated in Fig. S4 of Supplementary information. Similarly, there was also a reduction during the evening peak, up to -1.5 µg m−3 (-12%). Note that during the lockdown, the VOC emissions increased up to 7% in the stationary combustion sector (see Figure 3). Changes in emissions that showed a significant decrease in NO2 concentrations and slight decreases in VOC concentrations enhanced O3 levels over the AMB. This is consistent with the observations, where there was a decrease in NO2 concentrations (40-80%) and an increase in O3 levels (up to 10%) between 2015-2019 and 2020 during the lockdown (see Figs. S1-S2 in the Supplement). Note that, we need to consider the influence of the meteorological conditions to analyze changes in the air quality observations. The reduction of O3 concentrations that normally result from lower levels of precursors was canceled by a reduction in NO titration, resulting in a net increase in O3 levels. During the evening peaks (19-21 UTC), we found the highest increases in O3 of the COVID simulation compared to the BAU (1 to 18 µg m−3 , 1 to 20%). However, when surface O3 concentrations were higher (afternoon peak, 13-15 UTC), the increases in O3 levels were much lower (up to 6 µg m−3, 6%) than those for the evening peak (see Fig. S5 in Supplement). Outside the AMB, the concentrations did not differ significantly (< 2 µg m−3, < 2%) between BAU and COVID simulations. Differences in the Ox (NO2 + O3) values were calculated to aid our interpretation of the O3 concentrations by diminishing the effect of O3 titration by NO in highly polluted areas (see Fig. S6 in Supplement). The overall changes between BAU and COVID in the Ox concentrations remained practically constant due to a balance between the increases in O3 levels and decreases in NO2 levels. This has important policy implications because one air pollutant problem is being replaced by another, which is an undesirable consequence due to the ground-level ozone effects on human health, vegetation, and ecosystems. A similar result was seen by \cite{acp-21-4169-2021}, which found that Ox concentrations only changed very slightly due to the lockdowns across most European urban areas.

The differences between the BAU and COVID simulations for the second period (18 to 30 May) showed overall reductions in the NO2 (-2 to -15 µg −3, -10 to -65%) and VOC levels (up to -2 µg m−3, -16%), with high reductions found during the evening peaks (see Fig. S3 in the Supplement). Ozone levels decreased (by up to 3.5 µg m−3 , see Fig. S5 in the Supplement) in most of Catalonia due to significant reductions in most of the emission sectors (see Figure 3) during the COVID simulation, which decreased the high ozone productivity normally seen for this time of the year. However, we still found enhanced O3 levels around the Barcelona

airport in the evenings; the reductions in emission levels were still significant (more than 80%, see Figure 3) and inhibited titration of the O3 by NO. Note that in this case, the Ox concentrations decreased nearly everywhere in the Catalonia area and up to -4 µg m−3 over the AMB (see Fig. S6 in Supplement) for the COVID simulation, resulting in overall improvements in the air quality.

L27 is 1% significant?

Response: Thank you for pointing this out. We agree that 1% is not significant. This sentence have been removed from the manuscript.

L285 the 'two simulations' of what?

Response: Thank you. We have clarified in the updated manuscript that the two simulations are BAU and COVID. The text now reads:

Outside the AMB, the concentrations did not differ significantly (< 2 µg m−3, < 2%) for the two between BAU and COVID simulations.

L287 constant between what?

Response: Thank you. We mean constant between the two simulations analised: BAU and COVID. Now it is clarified on the text:

The overall changes between BAU and COVID in the Ox concentrations remained practically constant due to a balance between the increases in O3 levels and decreases in NO2 levels.

SS5.1 is interesting. I presume the graphs shown are for model results, and it would help to have this stated. Did the authors consider performing a similar analysis for observational data for this period? Is it difficult due to a lack of VOC data? The captions of Figure 6-8 needs to state explicitly that these are 'Changes...' between BAU and COVID

Response: Thank you for this suggestion. It is not possible to perform a similar analysis for observational data due to the lack of VOC data in this region. Figures 6-8 show ozone concentrations for the two model simulations: BAU and COVID. The captions of Figure 6-8 have been updated to make this clear.

Caption Figure 6: Modelled O3 concentrations (top panels) for 30 March to 12 April (only weekdays) and 18 to 30 May (only weekdays) for both simulations, BAU (left panels) and COVID (right panels), over the AMB area during the morning (6-8 UTC). Each dot of the top

row corresponds to the O3 concentration difference (ppb) of one grid cell of the AMB at the surface level. The dots in the lower row represent the land use for each model grid cell.

Colouring data by ozone change and land use/land cover is interesting, and the broken lines make the analysis goal clear. I would like to see the analysis better justified, though. I assume it is correct to use the lines which are derived from an analysis of transition regimes based on NOx and VOC emissions in Sillman (rather than changes in NOx/VOC levels used here) but I'd like the paper to discuss somewhere how these regimes apply when discussing a _change_ in O3 and a change in NOx or VOC levels, particularly in identifying regions of the diagrams here with NOx- or VOC-limited regimes, which seems key. I've not seen an analysis like this before, so would like to see this expanded upon.

Response: Thank you for this  constructive comment. NOx/VOCs regimes based on NOx/VOC levels are used in several papers such as Yang et al., (2021), Wang et al., (2021) and Ren et al., (2022). Yang et al., (2021) use the same lines and transition regimes as NOx/VOC emissions in Sillman et al., 1990.  Following Wang et al., (2021) and Ren et al., (2022) we establish a relationship between surface VOC/NOx and O3 concentrations, and, subsequently, we derive the line separating two different photochemical regimes by the local O3 maximum:

[Figure]

Figure S7: O3 concentration as a function of VOC/NOx concentration. Points are calculated with a 5 hours average concentration for the two periods using the BAU simulation. Dark vertical line separate the two photochemical regimes by the local O3 maximum. Grey vertical lines separate the transitional regimes.

The local O3 maximum occurs when VOC:NOx ≈ 8, coinciding with the ratio defined in Sillman et al., 1999. Therefore, we use the same lines as Silmann et al., (1999). We have added this Figure in the Supplementary information and add more information into the discussion to describe this analysis including new references.

Yang, L., Yuan, Z., Luo, H., Wang, Y., Xu, Y., Duan, Y., & Fu, Q. (2021). Identification of long-term evolution of ozone sensitivity to precursors based on two-dimensional mutual verification. In Science of The Total Environment (Vol. 760, p. 143401). Elsevier BV. https://doi.org/10.1016/j.scitotenv.2020.143401

Wang, W., van der A, R., Ding, J., van Weele, M., and Cheng, T.: Spatial and temporal changes of the ozone sensitivity in China based on satellite and ground-based observations, Atmos. Chem. Phys., 21, 7253–7269, https://doi.org/10.5194/acp-21-7253-2021, 2021.

Ren, J., Guo, F., and Xie, S.: Diagnosing ozone–NO$x$–VOC sensitivity and revealing causes of ozone increases in China based on 2013–2021 satellite retrievals, Atmos. Chem. Phys., 22, 15035–15047, https://doi.org/10.5194/acp-22-15035-2022, 2022.

Sillman, S.: The relation between ozone, NOx and hydrocarbons in urban and polluted rural environments, Atmospheric Environment, 33, 1821–1845, https://doi.org/https://doi.org/10.1016/S1352-2310(98)00345-8, 1999.

Section 5.1, lines 313-317:  It should be noted that Sillman et al., (1999) use changes in VOC:NOx emissions rather than changes in VOC:NOx levels used in this study. In this study, we follow Wang et al., (2021) and Ren et al., (2022) that establish a relationship between surface VOC:NOx and O3 concentrations, and, subsequently, derive the line separating the two different photochemical regimes by the local O3 maximum (see Figure S7 in the Supplement). The local O3 maximum occurs when VOC:NOx ≈ 8, coinciding with the ratio defined in Silmann (2003).

L315 and on, could the authors explain how the figures can be used to support this statement?

Response: Thank you for point this out. We have rewritten the text to clarify that in line 315 we are talking about "urban forest":

Overall, without any reduction in emissions (BAU simulation), this analysis indicates that urban forests far from anthropogenic sources and influenced by high biogenic VOC emissions, the photochemical regime of O3 formation is NOx-sensitive in the mornings and afternoons.

Following previous discussion, Figures 6 and 7 and Table 3 support that the photochemical regime of O3 formation is NOx-limited in the mornings and afternoons:

[Figure]

Figure 6: *Change in O3 concentrations (top panels) for 30 March to 12 April (only weekdays) and 18 to 30 May (only weekdays) for both simulations, BAU (left panels) and COVID (right panels), over the AMB area during the morning (6-8 UTC). The land use is also displayed for each grid (bottom panels).*

[Figure]

Figure 7: *Same as Figure 6 during the afternoon (13-15 UTC).*

Table 3. *Averages NOx /VOC ratio and ozone concentrations from 30 March to 12 April (only weekdays) and 18 to 30 May (only weekdays) in the morning (6-8 UTC), afternoon (13-15 UTC) and evening (19-21 UTC). Light grey and dark grey cells indicate VOCs and NOx regimes, respectively. The relative changes in ozone concentrations (%) are shown in brackets and were calculated as ((COVID-BAU)/BAU)×100.*

| March-April | Landuse | Morning | | Afternoon | | Evening | |
|---|---|---|---|---|---|---|---|
| | | BAU | COVID | BAU | COVID | BAU | COVID |
| $[NO_x/VOC]$ | Forest | 0.102 | 0.083 | 0.073 | 0.059 | 0.143 | 0.111 |
| | Natural Open | 0.158 | 0.098 | 0.109 | 0.077 | 0.194 | 0.122 |
| | Agriculture | 0.156 | 0.100 | 0.113 | 0.077 | 0.195 | 0.101 |
| | Water | 0.25 | 0.201 | 0.495 | 0.393 | 0.707 | 0.535 |
| | Compact urban | 0.166 | 0.139 | 0.171 | 0.137 | 0.236 | 0.184 |
| | Open urban | 0.125 | 0.088 | 0.108 | 0.081 | 0.182 | 0.135 |
| | Industrial | 0.149 | 0.120 | 0.135 | 0.100 | 0.215 | 0.151 |
| $O_3$ (ppb) | Forest | 40.3 | 40.7 (1.0 %) | 51.7 | 51.7 (0.0 %) | 42.3 | 42.9 (1.3 %) |
| | Natural Open | 42.8 | 43.9 (2.5 %) | 51.7 | 51.9 (0.3 %) | 45.4 | 46.4 (2.1 %) |
| | Agriculture | 37.3 | 38.5 (3.4 %) | 51 | 51.2 (0.4%) | 40.7 | 42.2 (3.7 %) |
| | Water | 37 | 38.7 (4.6 %) | 48.6 | 49.4 (1.6 %) | 42.1 | 44.2 (5.0 %) |
| | Compact urban | 35.8 | 36.9 (2.9 %) | 52 | 52.3 (0.6 %) | 40.4 | 42.2 (4.3 %) |
| | Open urban | 39.4 | 40.4 (2.4 %) | 52.6 | 52.8 (0.3 %) | 42.6 | 43.6 (2.4 %) |
| | Industrial | 36.8 | 37.7 (2.3 %) | 52.2 | 52.5 (0.5 %) | 40.1 | 41.8 (4.2 %) |

| May | Landuse | Morning | | Afternoon | | Evening | |
|---|---|---|---|---|---|---|---|
| | | BAU | COVID | BAU | COVID | BAU | COVID |
| $[NO_x/VOC]$ | Forest | 0.076 | 0.066 | 0.053 | 0.043 | 0.139 | 0.115 |
| | Natural Open | 0.122 | 0.087 | 0.078 | 0.052 | 0.194 | 0.129 |
| | Agriculture | 0.122 | 0.090 | 0.077 | 0.046 | 0.192 | 0.118 |
| | Water | 0.349 | 0.279 | 0.792 | 0.663 | 1.327 | 1.028 |
| | Compact urban | 0.137 | 0.121 | 0.156 | 0.125 | 0.261 | 0.210 |
| | Open urban | 0.101 | 0.078 | 0.099 | 0.072 | 0.191 | 0.150 |
| | Industrial | 0.125 | 0.104 | 0.107 | 0.075 | 0.230 | 0.165 |
| $O_3$ (ppb) | Forest | 36.9 | 36.7 (-0.6 %) | 51.3 | 50.5 (-1.6 %) | 37.7 | 37.6 (-0.1 %) |
| | Natural Open | 37.0 | 37.3 (0.6 %) | 48.9 | 48.3 (-1.3 %) | 40.3 | 40.6 (0.8 %) |
| | Agriculture | 33.3 | 33.8 (1.4 %) | 50.5 | 49.9 (-1.2 %) | 36.3 | 37.3 (3.0 %) |
| | Water | 29.7 | 31.2 (4.9 %) | 42.5 | 43.4 (2.2 %) | 32.0 | 35 (9.4 %) |
| | Compact urban | 33.6 | 33.9 (1.0 %) | 50.8 | 50.4 (-0.8 %) | 35.8 | 37.2 (3.8 %) |
| | Open urban | 36.7 | 36.9 (0.5 %) | 51.3 | 50.7 (-1.1 %) | 38.2 | 38.6 (1.1 %) |
| | Industrial | 34.3 | 34.6 (0.9 %) | 52.1 | 51.6 (-1.1 %) | 36.3 | 37.6 (3.5 %) |

L326, L330 the discussion reverts to ozone levels, not differences between scenarios. Could this discussion be made more consistent?

Response: Thank you for this comment. We have updated this part and now the discussion in Section 5.1 is more consistent and reads:

In terms of ozone levels (BAU simulation), high values during the morning (40-43 ppb) and evening (46-47 ppb) hours are found in suburban areas (forest and natural open) because there is less NO (because of less traffic) and thus less ozone degradation. In the afternoon, high O3 levels are found everywhere in the AMB (49-55 ppb), especially in urban areas (industrial, open urban and compact urban).

L319 grid points not grid.

Response: Amended.

SS5.2 discusses the oxidising capacity in terms of OH and NO3. Is there any impact on ozone budgets seen from changing HO (and presumably OH:HO2 ratio), particularly in HO2 + NO vs OH + NO2 vs OH + O3?

Response: Thank you for this constructive comment. The atmospheric oxidation capacity is to a large extent determined by budgets of the hydroxyl radical (OH). At night, the oxidizing capacity is due to the oxidation by NO3 and O3. Our discussion in the atmospheric oxidation capacity is based on the mixing ratios of OH, NO3 and O3. Making an enhaustive analysis of the ozone loss (O3+OH) and production (HO2+NO, OH+NO2) rates is out of the scope of this paper. However, we have now expanded the description of the oxidation capacity of the Introduction and re-written section 5.2. and now reads:

Introduction, lines 85-94: Only a few studies have reported enhanced atmospheric oxidation capacity (AOC), which describe the removal rate of primary pollutants and the formation of secondary species, and associated O3 increases during the COVID-19 lockdown due to increases in the major oxidants OH, hydroperoxy radical (HO2) and nitrate radical (NO3) (Zhu et al., 2021; Wang et al., 2021b, 2022). . The predominant oxidant for AOC during the daytime is OH, since NO3 radicals photolyse rapidly during daytime, which is responsible for the oxidation and removal of most natural and anthropogenic trace gases (Elshorbany et al., 2009; Saiz-Lopez et al., 2017). On the other hand, during the night the concentration of OH is significantly reduced, and the AOC is then controlled by NO3, together with O3, which is also an important oxidant  (Elshorbany et al., 2009; Saiz-Lopez et al., 2017). During the lockdown, the significant decreases in NO2 concentrations increased the OH levels, which led to the formation of harmful oxidants such as O3  (Zhu et al., 2021; Wang et al., 2021b, 2022).. Therefore, the AOC is an indicator used to design control policies for secondary species.

Section 5.2, lines 360-378: The increase of these free radical concentrations could be the leading cause for the diurnal O3 increases (see Fig. S5 in the Supplement) given that VOC and CO oxidation by OH are the initial reactions for ozone formation. In addition, the NO3 radical, which is a primary night-time oxidant, also increases in areas close to the airport and harbour (4 ppt, 210%). This increase can be explained by reductions in the VOC and NO2 levels, which are important sinks for NO3 radicals (Elshorbany et al., 2009; Saiz-Lopez et al., 2017).

During the period in May, we also found increases in the oxidants radicals (OH, NO3) and also O3 (see Fig. S5 in the Supplement) of the COVID simulation with respect to the BAU, in areas where substantial NOx emission reduction took place such as the airport and harbour. In these areas, OH levels increase up to 0.3 ppt (55%) in the afternoon, and NO3 increases up to 4 ppt (230%) in the evening. However, other areas showed general decreases in AOC radicals (-0.1 ppt for OH and -2 ppt for NO3), resulting in decreases in the O3 levels. Note that for both periods, the decrease in shipping emissions (a source of NOx) led to increases in the levels of both radicals along the ship tracks.

Our results indicate that changes in the anthropogenic emissions (mainly NOx and VOC) lead to significant changes in the OH and NO3 radicals levels, which in the case of emission reduction, such as that experienced during the COVID lockdown, lead to enhanced oxidation efficiency in the urban atmosphere of the AMB and O3 enhancements. However, during the period in May, when O3 formation increased due to warm temperatures and increase in biogenic emissions, there was a decrease in the AOC (except in the airport and harbour areas) for the COVID run. The elevations of AOC occurred because these areas were still VOC-limited regimes during this period. In terms of air quality policy, it is important to understand the interplay between these free radicals and O3 chemistry so that mitigation strategies are not counterproductive.

SS5.3 is very nice, and might be improved in consistency with a discussion of Ox (and maybe formation of Ox/NOx reservoirs and sinks) as in previous sections.

Response: Thank you for this comment. We have added more discussion of Ox in this section.

Section 5.3, lines 418-420: The decreases in ozone precursor emissions (COVID simulation) resulted in less ozone production from the AMB plume as well as production of new O3, and consequently, the ozone concentrations decreased.

Section 5.3, lines 428-430: The improvement in the air quality is consistent with the decrease in surface Ox concentrations seen during the period of May over most of the Catalonia region (see section 4) .

S6 summarises nicely. The sentence in L438 'this was consistent...' needs to be expanded.

Response: Thank you for this comment. We have now expanded this sentence and now reads:

This was consistent with the unchanged or decreased AOCs, given their relation with O3 production.

L454 'data used in this study...' add 'are'

Response: Amended.

Overall

Response: We do not understand this comment.

---

## Author Response (AR2)

**Response to the Editor and the Reviewer 2**

We thank the editor and reviewers for the constructive comments and suggestions which have helped us improve the manuscript. Our response is in blue, to differentiate from the comment, which is in black. Furthermore, we include any new text added in the manuscript in red, to facilitate this second revision.

**Editor comment:**

Notification to the authors:

1. Regarding the figure 1: with the next revision, please add the copyright icon as follows: © Google Earth. 2. Please remove the placeholder text from page 2 of the manuscript. 3. Please ensure that the colour schemes used in your maps and charts allow readers with colour vision deficiencies to correctly interpret your findings. Please check your figures using the Coblis – Color Blindness Simulator (https://www.color-blindness.com/coblis-color-blindness-simulator/) and revise the colour schemes accordingly.

**Response:** We thank the editor for his/her comments. The following changes have been done:

1. The copyright icon © has been added to Figure 1. Now, the caption from Figure 1 is: a) Location and b) main topographic features of the study area. Base maps in Panel a were taken from © Google Earth. The locations of air pollution monitoring stations (Xarxa de Vigilància i Previsió de la Contaminació Atmosfèrica, XVPCA) along the S–N axis (Barcelona-Vic Plain-Pyrenean range) are shown in Panel a (right).

2. The placeholder "TEXT" from page 2 has been removed.

3. All figures in the manuscript and the supplement has been checked using the Cobis - Color Blindness Simulator.

**Reviewer comment:**

"In their response to section, Section 5.1, lines 313-317, I find no mention in Sillman et al. 2003 of the values of the NOX:VOC ratios that they say are in there. If the authors can clarify where these numbers come from, that is to say why *precisely* the values of 4, 8 and 15 pertain, or give a reference in which these exact values are specified, I would be prepared to accept publication..."

**Response:** We thank the reviewer for this comment. The values of 4, 8 and 15 are not from Sillman et al. 2003, these values are from National Research Council (1991). In addition, in our study we follow other studies (e.g. Yang et al. 2021 and Ren et al. 2022) that establish a

relationship between surface VOC:NOx and O3 concentrations, and, subsequently, derive the line separating the two different photochemical regimes by the local O3 maximum. The local O3 maximum occurs when VOC:NOx ≈ 8, coinciding with the ratio defined in National Research Council (1991). Now it is clarified on the text:

Section 5.1, lines 307-314: In this study, we establish a relationship between surface VOC:NOx and O3 concentrations, and, subsequently, derive the line separating the two different photochemical regimes by the local O3 maximum (see Figure S7 in the Supplement). The local O3 maximum occurs when VOC:NOx ≈ 8, coinciding with the ratio defined in National Research Council (1991). In Figure 4-6, we indicate the NOx -limited regime with a dark solid line separating VOC:NOx >8, which is typical for locations located downwind of urban and suburban areas, and the VOC-limited regime (VOC:NOx <8) which is typical for highly polluted urban areas (National Research Council, 1991). We also indicate the transitional regime with two dotted lines (VOC:NOx >4/1 and VOC:NOx <15/1) showing where ozone becomes less sensitive to NOx changes and increases with increasing VOC levels, as identified in National Research Council (1991).

References:
National Research Council (NRC), 1991. Rethinking the Ozone Problem in Urban and Regional Air Pollution. Washington, DC: The National Academies Press. https://doi.org/10.17226/1889.